# Antibacterial and Synergistic Effects of *Terminalia citrina* Leaf Extracts Against Gastrointestinal Pathogens: Insights from Metabolomic Analysis

**DOI:** 10.3390/antibiotics14060593

**Published:** 2025-06-08

**Authors:** Sze-Tieng Ang, Tak Hyun Kim, Matthew James Cheesman, Ian Edwin Cock

**Affiliations:** 1School of Environmental and Science, Griffith University, Nathan Campus, Brisbane 4111, Australia; k.ang@griffith.edu.au (S.-T.A.); t.kim@griffith.edu.au (T.H.K.); 2School of Pharmacy and Medical Sciences, Griffith University, Gold Coast Campus, Gold Coast 4222, Australia

**Keywords:** Combretaceae, plant extracts, gastrointestinal pathogens, MRSA, antibacterial activity, combinational therapies, metabolomics

## Abstract

**Background/Objectives**: Bacterial contamination leads to foodborne illnesses, and new antibiotics are required to combat these pathogens. Interest has increased in medicinal plants as targets for new antibiotics. **Methods**: This study evaluated the antibacterial activity of leaf extracts from *Terminalia citrina* (Gaertn.) Roxb. ex Fleming against four bacterial pathogens (including a methicillin-resistant *Staphylococcus aureus* (MRSA) strain) using disc diffusion and liquid microdilution assays. The phytochemical composition of the extracts were determined using ultra-high-performance liquid chromatography–mass spectrometry (UPLC-MS). **Results**: Both the aqueous and methanol extracts demonstrated noteworthy antibacterial activity against *Bacillus cereus* (MICs of 468.8 µg/mL and 562.5 µg/mL, respectively). Additionally, the extracts were effective against MRSA (MICs = 625 µg/mL). Strong antibacterial effects were also observed against *S. aureus*, with MICs of 625 µg/mL (aqueous extract) and 833.3 µg/mL (methanol extract). Twelve combinations of extracts and conventional antibiotics were synergistic against *B. cereus* and *S. flexneri*. UPLC-MS analysis revealed two flavonoids, orientin 2″-O-gallate and astragalin, exclusive to the aqueous extract, whilst pinocembrin and gallic acid were only detected in the methanol extract. Both extracts contained vitexin 2″-O-p-coumarate, ellagic acid, orientin, rutin, chebulic acid, terminalin, and quercetin-3β-D-glucoside. Both extracts were determined to be nontoxic. **Conclusions**: The abundance and diversity of polyphenols in the extracts may contribute to their strong antibacterial properties. Further research is required to investigate the antibacterial effects of the individual extract compounds, including their effects when combined with conventional antibiotics, and the potential mechanisms of action against foodborne pathogens.

## 1. Introduction

Foodborne diseases cause significant suffering and loss of life, affecting nearly one in ten people each year, with an estimated thirty-three million life years lost [1]. These illnesses often arise from bacterial contamination during manufacturing, storage, or food preparation, disrupting the digestive system’s microbial balance. Foodborne intoxication happens when toxins produced by bacteria are consumed, such as those from *Clostridium botulinum*, which can lead to severe illness if foods are improperly processed. *Staphylococcus aureus* has recently emerged as a major cause of food poisoning, especially in poultry and cooked meats, leading to what is known as ‘Staph food poisoning’ (SFP) [2]. Poultry and cooked meat products, such as ham or corned beef, are identified as the most SFP-impacted foods [2]. Notable outbreaks include one in 2015 linked to Chantilly cream in Italy, leading to twenty-four illnesses [3], and another in 2019 from improperly handled chicken salad, resulting in eleven hospitalizations [4]. Symptoms of SFP generally appear within one to six hours and include nausea, vomiting, and abdominal pain.

The rise of antibiotic-resistant bacteria, such as methicillin-resistant (MRSA), is making infections harder to treat. A large-scale SFP outbreak at a school canteen in Vietnam in 2018 identified that the *S. aureus* strain responsible for the outbreak was also resistant to penicillin and tetracycline [5]. In addition, MRSA strains with a high level of resistance towards amikacin and amoxicillin–clavulanate were detected in shellfish samples from retail markets in 2017 in Malaysia [6]. Furthermore, the Ministry of Health in Malaysia reported in 2017 that there had been an increased number of MRSA strains detected in clinical isolates, as well as an increasing trend of strains resistant to penicillin, clindamycin, and erythromycin [7]. This trend of increasing antibiotic resistance is concerning, as there are fewer effective antibiotics to treat bacterial infections.

Shigellosis, caused by the *Shigella* bacteria, was the second leading cause of diarrheal mortality in 2016, resulting in roughly 212,000 deaths worldwide [8]. A recent report by the Centre for Disease Control and Prevention (CDC) noted a rise in cases due to extensively drug-resistant (XDR) *Shigella*, from 0% in 2015 to 5% in 2022 [9]. *Bacillus cereus* is another prevalent foodborne bacterium known for its resistance to extreme temperatures, allowing it to survive drying, heating, and cold storage. Indeed, Thery et al. (2022) reported a case of an 11-year-old child who suffered from multiple organ failure after consuming toxins produced by *B. cereus* in pasta cooked three days earlier [10]. The strain involved was resistant to several antibiotics, and even after treatment with intravenous vancomycin, it remained in the child’s stool samples after five days.

The main treatment for bacterial gastrointestinal (GI) diseases is antibiotics. However, the rise in antibiotic-resistant bacteria has decreased the effectiveness of these therapies, creating significant health risks. This situation poses a critical challenge to global health, highlighting the urgent need for alternative treatment options. Despite efforts to combat antibiotic resistance, new antibiotic discoveries have declined since the late 1960s due to high development costs, lengthy trial timelines, and low profitability. From 2003 to 2023, only 33 new antibacterial drugs were approved, most of which are narrow-spectrum and limited in their effectiveness [11,12]. Repurposing conventional antibiotics to restore their antibacterial activities by adding potentiators is attracting substantial recent attention [13,14,15].

The use of plants in traditional healing systems for treating bacterial infections dates back to ancient times and is grounded in indigenous beliefs and practices. The genus *Terminalia* (family Combretaceae) is recognized for its medicinal properties, with various species used globally to address diarrhea, inflammation, and bacterial infections [16]. *Terminalia* species are known for their high polyphenol content, which contributes to their antioxidant, antimicrobial, anti-inflammatory, and anti-cancer effects. Whilst substantially more work is required, several bioactive compounds have been highlighted in previous studies. For example, chebulinic acid extracted from *T. chebula* Retz. fruit extracts has strong antibacterial activity against an MDR clinical strain of *Acinetobacter baumannii* [17]. Furthermore, it also has significant anti-ulcer effects by inhibiting *Helicobacter pylori* growth [17]. Ou et al. (2024) showed that chebulinic acid acts as an anti-adhesive agent, hindering the ability of *H. pylori* to adhere to host cells, which is an important step in the progression of infection [18].

Several secondary metabolites derived from *Terminalia* species have demonstrated significant inhibitory activity against MDR bacterial strains, making them a promising avenue for antibiotic resistance research. Among these metabolites, compounds such as ellagic acid, gallic acid, flavonoids, and hydrolysable tannins, specifically punicalagin and terchebulin, have been particularly effective. These compounds have exhibited potent antibacterial properties against problematic MDR pathogens, including MRSA, *S. aureus*, and *Escherichia coli* [19,20]. Phytochemicals, based on their chemical structures and properties, can exhibit antibacterial actions through one or more mechanisms. For example, gallic acids induce irreversible changes in membrane properties, such as charge, permeability (both intra- and extracellular), and overall physicochemical characteristics [19]. These changes occur through alterations in hydrophobicity, a decrease in negative surface charge, and the formation of local ruptures or pores in the cell membranes, leading to the leakage of essential intracellular components. Additionally, phytochemicals may interfere with microbial resistance pathways, undermining the bacteria’s strategies to evade antibiotic effects [20].

Despite their widespread distribution across tropical and subtropical regions, many *Terminalia* species remain underexplored for their antibacterial potential. *Terminalia citrina* (Gaertn.) Roxb. ex. Fleming is native to tropical regions of Asia. In the Philippines, fruit decoctions are used to treat thrush and persistent diarrhea, whilst in Indonesia, decoctions are commonly used to treat gastrointestinal illnesses [21]. Isolated tannins and extracts produced from *T. citrina* fruits inhibit the growth of several foodborne pathogens, including *B. cereus*, *S. aureus*, *E. coli*, *Klebsiella pneumoniae*, and several *Salmonella* and *Shigella* strains [22]. Despite this, no detailed phytochemical profiles exist, and previous studies have mainly focused on antibiotic-sensitive strains, highlighting the need for research on antibiotic-resistant bacteria. Furthermore, the interactions of these extracts with commercial antibiotics have not been adequately explored. This study aims to evaluate the antibacterial properties of *T. citrina* leaf extracts against gastrointestinal disease-related bacteria and to assess their combined effects with clinical antibiotics. The polyphenol profiles of the extracts are also analysed using ultra-high-performance liquid chromatography coupled with an Orbitrap Exploris 120 mass spectrometer (UPLC-MS).

## 2. Results

### 2.1. Extraction Yields and Antibacterial Activities

One-gram samples of dried powdered *T. citrina* leaves were extracted separately using sterile deionized water (TciW) and AR methanol (TciM). The dried extracted materials were resuspended in 10 mL of 1% DMSO, yielding concentrations of 21.58 mg/mL and 23.80 mg/mL, respectively.

### 2.2. Disc Diffusion Assay

The growth inhibitory activity of *T. citrina* leaf extracts was initially examined using disc diffusion assays on agar plates inoculated separately with *S. aureus*, MRSA, *B. cereus*, or *S. flexneri*. The zones of inhibition (ZOIs) were measured as a preliminary screening of the bacteria’s susceptibility to the treatment. Four antibiotics (cefotaxime 30 µg, ciprofloxacin 30 µg, tetracycline 10 µg, and vancomycin 5 µg) were tested concurrently on the agar plates as the positive controls. Notably, both *T. citrina* extracts inhibited the growth of all four bacterial strains, although with substantially lower efficacy than the positive controls (Figure 1). The analysis of the ZOI values demonstrated that both extracts were more effective inhibitors of the Gram-negative *S. flexneri* than the three Gram-positive strains.

### 2.3. Liquid Microdilution Assay

To quantify the growth inhibitory activity of the *T. citrina* leaf extracts, liquid microdilution assays were performed and expressed as minimum inhibitory concentration (MIC) values (µg/mL). This study classified extracts with MIC values >5000 µg/mL as inactive. Those with MIC values between 2000 and 5000 µg/mL were considered weak inhibitors, whilst MIC values between 1000 and 2000 µg/mL were regarded as having moderate activity. Values between 400 µg/mL and 1000 µg/mL were classified as noteworthy, while extracts with MIC values between 100 µg/mL and 400 µg/mL were classified as strong inhibitors. Finally, extracts with MIC values less than 100 µg/mL were deemed potent [23,24]. Both extracts (Table 1) inhibited the growth of the four tested bacteria, with the stronger antibacterial effects observed against *S. aureus*, MRSA, and *B. cereus*.

The methanol extract strongly inhibited the growth of MRSA, which aligns with the trend observed in the disc diffusion assay, which showed substantial zones of inhibition (Figure 1). Notably, the MRSA strain tested was resistant to methicillin and multiple other antibiotics, with an MIC that was approximately 25% higher than against the non-antibiotic-resistant strain *S. aureus*. Interestingly, the aqueous extract had greater antibacterial effects than the methanol extract towards *B. cereus* (as judged by MIC), and both extracts had similar growth inhibitory strengths against *S. flexneri*. Thus, both extracts were more potent against Gram-positive bacteria than against Gram-negative bacteria (based on MIC values). This observation contrasts with the disc diffusion results, as the extracts were more effective against the Gram-negative *S. flexneri* in the disc diffusion assay (Figure 1). The results obtained for the conventional antibiotics were consistent with those obtained from the disc diffusion assay. Both ciprofloxacin and tetracycline were effective toward all four tested bacteria at the tested concentration.

### 2.4. Fractional Inhibitory Concentration (FIC) Determination

The combinational effect of the *T. citrina* extracts and selected conventional antibiotics (cefotaxime, ciprofloxacin and tetracycline) was determined by ƩFIC analysis at a 50:50 ratio. ∑FIC values are used to classify the interactions between extract and antibiotic (Table 2) against the tested bacterial strains. Notably, no synergistic interactions were observed for any combination, although three additive interactions were noted in combinations containing cefotaxime, two in combinations containing ciprofloxacin, and five in combinations containing tetracycline. Additionally, no antagonistic interactions were observed at a 50:50 ratio. Therefore, these combinations do not decrease the activity of either component against the tested bacteria strains. The combination of vancomycin and the extracts showed indifference against the tested bacterial strains compared to the MIC alone.

### 2.5. Isobologram Analysis

The antibiotic/extract combinations that yielded synergy or additive effects were further evaluated across multiple ratios by plotting isobolograms (Figure 2 and Figure 3). Only synergistic and additive ratios are shown in these figures. Notably, both *T. citrina* extracts had synergistic effects against *B. cereus* when tested in combination with tetracycline at several combination ratios (Figure 2). A ratio containing 90% tetracycline and 10% aqueous extract was synergistic. All other ratios showed either additive or indifferent effects (Figure 2A). Synergistic effects were also observed when tetracycline was combined with the methanol extract for all combination ratios tested (Figure 2B) against *B. cereus*, except for combinations containing 50% extract (additive effect) or 60% extract (indifferent effect).

Another synergistic interaction was observed when *T. citrina* aqueous extract was combined with ciprofloxacin at a ratio of 70% of the extract against *S. flexneri* (Figure 3A). Other combination ratios only exhibited additive effects against *S. flexneri*. Additionally, combinations containing 70–90% of the aqueous extract combined with tetracycline exhibited synergistic effects against *S. flexneri* (Figure 3B). Additive effects were also observed for other ratios.

### 2.6. Quantification of Toxicity

*Artemia franciscana* (brine shrimp) lethality assays and human primary dermal fibroblast (HDF) cytotoxicity assays were used to evaluate the toxicity of the extracts. Notably, both extracts lacked toxicity at 500 and 1000 µg/mL, although they were toxic at 5000 µg/mL. As LC_50_ values >1000 µg/mL are defined as non-toxic in this assay, both extracts were deemed to be non-toxic. Potassium dichromate (1 mg/mL), which was used as the positive control, induced 100% mortality following 24 h of exposure, whilst the negative control (seawater) induced 0% mortality. In addition, the cytotoxicity assay using HDF indicated that both extracts were non-toxic. Exposure of the HDF to all extracts resulted in over 50% cell viability at 300 µg/mL.

### 2.7. UPLC-MS Polyphenol Fingerprinting Studies

The phytochemical profile of each extract was evaluated with an ultra-high liquid chromatography–Orbitrap Exploris 120 mass spectrometer (UPLC-MS), using optimised parameters previously developed in our group [27]. Phytochemical analysis of both aqueous and methanolic extracts was conducted using negative-ion and positive-ion modes. The total compound chromatograms for the negative-ion and positive-ion modes are displayed in Appendix A, respectively.

The polyphenol compounds identified in negative-ion mode ([M-H]-1) are reported herein, as this mode is generally regarded as superior for identifying polyphenols [28,29,30,31]. Polyphenol compounds identified in positive-ion mode ([M+H]+1) are included in Appendix A, and all other identified phytochemicals in both positive-ion and negative-ion modes are listed in Appendix A. Mass spectral analysis (Appendix A) resulted in the tentative identification of 35 polyphenol compounds (Table 3). The relative abundance of the identified compounds in the table corresponds to the percentage of the total area under the chromatographic peaks, and only compounds with more than 0.01% of the total area found in either of the extracts are documented herein. Compounds with less than 0.01% of the total area were considered trace amounts.

## 3. Discussion

Traditional medicines offer a promising avenue for discovering new antibiotics, with a notable increase in research on herbal therapies for infectious diseases globally. Many *Terminalia* species, especially in South Asian medicinal systems, are used in treating fever and gastrointestinal disorders [32,33]. Whilst previous studies have focused on the antibacterial properties of southern Asian species [33], research on *Terminalia citrina* is limited. Its high phenolic content and traditional medicinal use indicate potential for antibacterial drug discovery [34]. *Terminalia citrina* is valued in East and Southeast Asia (particularly in Ayurvedic medicine) for improving digestion and treating dysentery [35]. *Terminalia citrina* leaf and fruit extracts possess antimicrobial, antioxidant, and anti-diarrheal properties [36,37]. However, fruit extracts have demonstrated cytotoxicity against BHK-21 fibroblast cells and mild toxicity towards *Artemia* nauplii, highlighting the need for further investigation into the safety of *T. citrina* fruit extract [38,39].

Our study explored the antibacterial effects of *T. citrina* leaf extracts against GI bacterial pathogens. The disc diffusion assay showed that both aqueous and methanol extracts were effective against Gram-positive and Gram-negative bacteria, particularly the Gram-negative *S. flexneri* (Figure 1). However, a higher concentration (MIC value = 2500 µg/mL) was needed to inhibit *S. flexneri* growth in liquid media, whilst an MIC of ≤1000 µg/mL indicated significant antibacterial activity. The variability in results may be due to the limited mobility of the active compounds in solid media, as they may diffuse more slowly through agar gels, resulting in higher activity in the disc diffusion assay.

Both *T. citrina* extracts exhibited noteworthy antibacterial effects against Gram-positive bacteria, with MIC values between 400 and 1000 µg/mL (Table 1). The aqueous extract demonstrated greater inhibitory effects against *S. aureus* than the methanol extract, with a MIC of 625 µg/mL. Both extracts showed similar efficacy against antibiotic-sensitive *S. aureus* and the MRSA strain, suggesting different mechanisms of action compared to β-lactam antibiotics or that the extract components target antibiotic resistance pathways. Notably, extracts from other *Terminalia* species also demonstrate greater antibacterial activity against Gram-positive than Gram-negative bacteria [27,40,41]. However, most of the earlier studies focus on the disc diffusion assay without reporting MIC values, making comparisons difficult [36,42].

Variations in antibiotic resistance mechanisms among bacterial strains led to the selection of five antibiotics from different classes for the MIC assay. Two β-lactam antibiotics, methicillin and cefotaxime, were tested, although all strains showed β-lactam resistance, likely due to the production of β-lactamase enzymes that reduce the effectiveness of these drugs. In the case of MRSA, the primary resistance mechanism to β-lactam antibiotics is the production of an alternative penicillin-binding protein, PBP2a, which is encoded by the mecA gene. This results in a reduced affinity for antibiotics in this class [43]. Vancomycin, which belongs to the glycopeptide class of antibiotics, is one of the first-line treatments for MRSA infections. However, the frequent use of vancomycin as the primary option for these infections has already resulted in the emergence of MRSA isolates with reduced vancomycin susceptibility [44]. Notably, the MRSA strain tested herein displayed a notable reduction in susceptibility to vancomycin, with an MIC of 3.8 µg/mL. This measurement clearly positions it within the intermediate susceptibility range as defined by Clinical and Laboratory Standards Institute (CLSI) guidelines, highlighting the challenges in treating infections caused by this pathogen. That strain also exhibited reduced susceptibility to other β-lactam, glycopeptide, and fluoroquinolone antibiotics. This situation underscores the urgent need for new therapeutic drugs and treatment strategies.

Tetracycline was the only conventional antibiotic tested in this study that effectively inhibited MRSA. This antibiotic targets protein synthesis to inhibit bacterial growth [45]. Similarly, the tested *B. cereus* strains were susceptible to tetracycline, ciprofloxacin, and vancomycin, although it was resistant to cefotaxime and methicillin. These findings suggest that the resistance profile of *B. cereus* in this study may be limited to β-lactam antibiotics that target cell wall synthesis. Ciprofloxacin is a broad-spectrum antibiotic in the fluoroquinolone class that inhibits bacterial DNA gyrase and topoisomerase IV, resulting in cell death [46]. Both *S. aureus* and *B. cereus* strains demonstrated susceptibility to ciprofloxacin. In contrast, the *S. flexneri* strain tested in this study exhibited an elevated MIC that exceeded the CLSI susceptibility breakpoints [25]. The slightly elevated MIC observed for *S. flexneri* ATCC 12022 (2.5 µg/mL) may be attributed to experimental variability, such as differences in inoculum density or growth medium conditions and does not necessarily indicate acquired resistance. Importantly, the MIC for tetracycline against *S. flexneri* (2.5 µg/mL) remains well within the susceptible range (with the CLSI resistance breakpoint set at ≥ 16 µg/mL) [25], further supporting the reliability of the data obtained. These results indicate that, despite some resistance to β-lactam antibiotics, all bacterial strains tested remained sensitive to drugs that target the cell wall through various mechanisms. Additionally, they consistently showed susceptibility to antibiotics that disrupt protein production and DNA replication. This suggests that these classes of antibiotics may still be effective options for treating infections caused by these bacteria.

Combinational studies involving antibiotics and extracts were conducted to explore potential interactions between these substances. Notably, the 50:50 combination of the aqueous extract and cefotaxime had an additive effect when tested against *S. aureus* (Table 2). Therefore, phytochemicals in the extract may inhibit β-lactamase activity. The methanol extract and tetracycline combination also exhibited additive effects against *S. aureus*, MRSA, and *B. cereus*. The combination of aqueous extract and tetracycline also showed additive interaction against *B. cereus* and *S. flexneri*. As tetracycline resistance is primarily attributed to tetracycline-specific efflux pumps, the enhanced antibacterial strength in these combinations may indicate that the extract components reduce/block the efficacy of these efflux pumps. As a result, tetracycline may remain at higher concentrations inside the cell longer, thereby hindering the protein translation process. Phytochemicals in the methanol extract enhanced the antibacterial activity of tetracycline against Gram-positive bacteria. Similarly, the phytochemicals in the aqueous extract improved the antibacterial effectiveness when combined with tetracycline. Further investigation is needed to evaluate the mechanisms underlying these interactions.

Additionally, two other additive interactions were observed when the aqueous extract was combined with ciprofloxacin and tested against MRSA and *S. flexneri*. Ciprofloxacin is a bactericidal antibiotic that inhibits bacterial DNA replication. The aqueous extract may contain compounds that can interfere with the bacterial resistance mechanisms to ciprofloxacin, which include mutations in target proteins and the production of ciprofloxacin-specific efflux pumps. No synergistic or antagonistic interactions were observed at other 50:50 combinations. However, synergy was noted when the aqueous extract was combined with tetracycline at three different ratios, as well as with 70% ciprofloxacin in tests against *S. flexneri* (Figure 3). The aqueous extract may harbour active compounds that hinder the resistance mechanisms of ciprofloxacin, including mutations in target proteins and the production of efflux pumps. Notably, various combinations of the aqueous extract with 30% or less of tetracycline and ciprofloxacin demonstrated a synergistic effect against *S. flexneri*. This indicates that the active compounds in the aqueous extract may act as competitive inhibitors for the antibiotic-specific efflux pumps, thereby enhancing the likelihood of the antibiotics penetrating the bacterial cells and effectively inhibiting their growth.

Based on these findings, it appears that both extracts may contain phytochemicals that interfere with the bacterial resistance mechanisms against ciprofloxacin and tetracycline. However, further studies are necessary to confirm this. All other combination ratios tested exhibited additive or non-interactive effects, suggesting that they are safe for simultaneous use, providing additional or no added advantages. To ensure the safe use of the extracts, toxicity tests have been carried out using ALA lethality assay and MTS cellular viability assay. Both extracts were confirmed to be non-toxic in each assay model.

UPLC-MS was used in this study to identify thirty-five polyphenolic compounds (relative abundance >0.01%) in both extracts, which include twenty-one flavonoids, thirteen tannins, and one lignan compound (Table 3). A complete and comprehensive list of all the phytochemicals identified in each extract is available in the Appendix A. Both extracts contained thirty-five polyphenolic compounds, with two flavonoid compounds exclusive to the aqueous extract and another two polyphenol compounds unique to the methanol extract. The flavonoids that were exclusive to the aqueous extract were astragalin (Figure 4A) and orientin 2″-O-gallate (Figure 4B). Gallic acid (Figure 4C) and pinocembrin (Figure 4D) were exclusive to the methanol extract.

Other noteworthy polyphenol compounds identified in both extracts include catechin (Figure 5A), ellagic acid (Figure 5B), orientin (Figure 5C), (-)-epicatechin gallate (Figure 5D), vitexin 2″-*O*-*p*-coumarate (Figure 5E), chebulic acid (Figure 5F), formononetin (Figure 5G), luteolin (Figure 5H), (-)-epigallocatechin gallate (Figure 5I), rutin (Figure 5J), procyanidin B3 3-*O*-gallate (Figure 5K), quercetin-3β-D-glucoside (Figure 5L), cinnamtannin A4 (Figure 5M), terminalin (Figure 5N), quercetin 3-(6″-*p*-hydroxybenzoylgalactoside) (Figure 5O), 6-*O*-galloyl-glucose (Figure 5P), 1,3,6-trigalloyl glucose (Figure 5Q), and corilagin (Figure 5R). Catechin, ellagic acid, gallic acid, quercetin-3β-D-glucoside, rutin, luteolin, terminalin, 6-*O*-galloyl-glucose, 1,3,6-trigalloyl glucose, corilagin, and chebulic acid were previously reported to be found in many *Terminalia* species, including fruit extract of *T. citrina* [27,39,47].

Flavonoids are known for their antibacterial properties [48]. Catechins, a subgroup of flavonoids known as flavan-3-ols, exhibit strong antibacterial properties [49]. Notably, both of the *T. citrina* extracts tested in our study were rich in catechins. Indeed, the catechins and their derivatives, including (-)-epicatechin gallate, (-)-epigallocatechin gallate (EGCG), epigallocatechin 3-*O*-(4-hydroxybenzoate), cinnamtannin A3 (polymers of catechins), cinnamtannin A4 (polymers of catechins), and procyanidin B3 3-*O*-gallate (polymer of catechin and gallic acid), were the primary polyphenols identified in both extracts.

Several publications have also reported the presence of catechins in fruit and bark extracts of *Terminalia* species [50]. Friedman et al. (2005) reported the antibacterial activity of seven green tea catechins against *B. cereus*, with the most active catechin, (-)-gallocatechin-3-gallate, exhibiting 28-fold and 44-fold lower concentrations compared to tetracycline and vancomycin, respectively [51]. Additionally, Osterburg et al. (2009) reported that EGCG from green tea (*Camellia sinensis* (L.) Kuntze) had MIC ranging from 78 to 625 µg/mL against 21 clinical isolates of *Acinetobacter baumannii* [52]. The combination of EGCG and the topical agent 5% mafenide acetate (Sulfamylon) demonstrated a powerful synergistic effect against a clinical bacterial isolate of *A. baumannii*. [52]. This observation demonstrates the potential of individual catechin compounds to enhance the antibacterial activity of conventional antibiotics. The differences in binding affinity of EGCG to the cell wall components of Gram-positive and Gram-negative bacterial strains were also reported by Yoda et al. (2004), and this variation greatly affected their susceptibility to EGCG [53]. These findings are similar to those collected in our study, as both catechin-rich *T. citrina* extracts demonstrated stronger MICs against Gram-positive bacterial strains compared to Gram-negative strains. A report by Buchmann et al. (2021) demonstrated that combining EGCG with β-lactam antibiotics has synergistic effects against *S. aureus* and MRSA [51,54]. This synergy may be attributed to the direct binding of EGCG to the peptidoglycan in the bacterial cell wall, potentially compromising its integrity. As a result, this interaction may enhance the penetration of small molecules, such as antibiotics, leading to more effective antibacterial action. This finding lays the groundwork for future research, including studies on various β-lactam antibiotics to enhance their effectiveness.

Interestingly, the gallic acid monomer was only detected (using both positive-ion ([M+H]+1) and negative-ion ([M-H]-1) modes) in the methanol extract of *T. citrina* leaves. Although the aqueous extract lacked detectable levels of gallic acid monomer, both extracts contained notable amounts of gallic acid derivatives, with the methanol extract having a greater relative abundance than the aqueous extract. Three gallic acid derivatives were isolated from the extracts in addition to the previously discussed catechin–gallic acid derivatives. These derivatives include 6-*O*-galloyl-glucose, 1,3,6-trigalloyl glucose, and corilagin. Gallic acid compounds can induce damage to both the outer and inner membranes of *E. coli* by reducing the mRNA expression levels of the genes *arcA*, *arcB*, *arcD*, *arcF*, and *tolC* [55]. As these genes are associated with membrane permeability, it is likely that the gallic acid derivative functions via changes in membrane permeability [55]. That study also reported that gallic acid inhibits biofilm formation in *E. coli* and has a synergistic effect when used in combination with tetracycline and ceftiofur against selected *E. coli* strains [55]. Similarly, Hossain et al. (2020) reported that the combination of gallic acid and ceftiofur exhibited an additive interaction by inhibiting the formation of *Salmonella* Typhimurium biofilms [56]. Additionally, galloyl glucose derivatives have antibacterial effects against both Gram-positive and Gram-negative bacteria, with Gram-positive bacterial strains showing greater susceptibility [57,58]. A study by Ou et al. (2024) found that 1,3,6-trigalloyl glucose, isolated from *Terminalia chebula* Retz, inhibits the growth of *Helicobacter pylori*, a Gram-negative bacterium that resides in the stomach lining and causes inflammation [59]. This compound damages the bacterial cell structure and suppresses the cytotoxin-associated gene A (Cag A) protein, which is responsible for the infection. However, that study did not investigate the interaction between 1,3,6-trigalloyl glucose and conventional antibiotics. In our study, the methanol extract of *T. citrina*, which contains notable levels of gallic acid, 1,3,6-trigalloyl glucose, and catechin derivatives, demonstrated synergistic interactions with tetracycline at various combination ratios when tested against *B. cereus*. In contrast, the combination of the aqueous extract and tetracycline showed no synergistic effects against *B. cereus*. These findings suggest that one or more of these compounds may help inhibit tetracycline resistance mechanisms. Such mechanisms may include blocking the protein expression associated with the tetracycline efflux pump or preventing the enzymatic inactivation of tetracycline.

Corilagin consists of three gallic acid molecules attached to a glucose molecule. Several studies have reported that corilagin suppresses the activity of β-lactamase enzymes in MRSA and therefore significantly lowers the MICs of β-lactam antibiotics when tested in combination, reducing them by a factor of 100 to 2000 times [60,61]. Notably, corilagin was previously identified in *T. chebula* extracts [62,63]. Interestingly, *T. chebula* fruit extract, as well as pure corilagin, inhibit biofilm formation when tested against *S. aureus* at 130 and 3.18 µg/mL, respectively [64]. Similarly, ellagic acid has promising antibiofilm inhibition properties. A recent study reported biofilm inhibitory activity for ellagic acid [65]. The interaction between ellagic acid and the enzyme responsible for biofilm formation in *E. coli*, NADH-quinone oxidoreductase WrbA, resulted in the inhibition of the WrbA enzyme [65]. Another study demonstrated that ellagic acid and its derivatives isolated from the root of *Rubus ulmifolius* Schott exhibit antibiofilm activity against *S. aureus* and enhance antibiotic susceptibility without toxicity to normal human mammalian cells [66]. The flavonoid vitexin 2″-*O*-*p*-coumarate is the second most abundant polyphenol identified in both extracts of *T. citrina*. Structurally, it consists of a vitexin molecule conjugated with a *p*-coumaric acid molecule. To the best of our knowledge, this is the first report documenting the presence of vitexin 2″-*O*-*p*-coumarate in a *Terminalia* species. Interestingly, vitexin 2″-*O*-*p*-coumarate isolated from the seeds of *Trigonella foenum-graecum* L. promotes the proliferation of human embryonic lung diploid fibroblast cells (2BS cells) under a H_2_O_2_ stress-induced environment, resulting in a viability rate of 25.44% [67]. The high concentration of this compound in *T. citrina* extracts (2.31% in aqueous extract and 0.55% in methanol extract) indicates that *T. citrina* may possess substantial antioxidant properties, although this remains to be verified.

The UPLC-MS experiments presented herein also identified several quercetin derivatives, including quercetin 3–6″-*p*-hydroxybenzoylgalactoside, quercetin-3β-D-glucoside (Q3G), and rutin. Quercetin and its derivatives have substantial antimicrobial, antioxidant, anticancer, anti-inflammatory, antidiabetic, anti-obesity, anti-Alzheimer, and antiviral activities [68,69,70]. Several other *Terminalia* species have also been reported to contain quercetin and its derivatives. Notably, extracts from those species have antibacterial activity against multiple bacteria, including some antibiotic-resistant strains. [27,40,47,71]. A methanol extract of *T. bellirica* (Gaertn.) Roxb. was reported to contain 0.33% (% relative abundance) quercetin and 0.14% rutin [27]. That study also reported that the extract strongly inhibited *S. aureus* growth, with an MIC of 94 µg/mL, and an MIC of 189 µg/mL against MRSA. It also had moderate antibacterial activity (MIC = 755 µg/mL) against *E. coli* and its corresponding extended spectrum β-lactamase (ESBL)-resistant *K. pneumoniae* strain. Additionally, quercetin has demonstrated antibacterial activity by binding to bacterial gyrase, an enzyme crucial for maintaining DNA structure and preventing overcoiling [72]. It disrupts the ATP binding site of gyrase, thereby inhibiting its function. This interference impairs the bacteria’s ability to replicate and transcribe their DNA, ultimately leading to cellular dysfunction and death. This mechanism is similar to that of ciprofloxacin, a conventional antibiotic tested in this study. Interestingly, synergistic interactions were observed when the aqueous extract of quercetin was used in conjunction with ciprofloxacin against S. flexneri at a specific combination ratio. This finding suggests that these phytochemicals may enhance the antibacterial effect of ciprofloxacin, possibly by inhibiting the active drug efflux pump and increasing the retention of ciprofloxacin within the cell or by exerting their antibacterial effects alongside ciprofloxacin at the same site.

Q3G also has several other reported bioactivities, including antiviral, antioxidant, and antimycobacterial effects [73,74,75]. However, reports of its activity against other bacterial pathogens are limited. Tomas-Menor et al. (2015) reported a synergistic effect between Q3G and other polyphenols (Q3G-ellagic acid; Q3G-myricetin; Q3G-punicalagin) isolated from *Cistus salviifolius* L. against *S. aureus* (CECT 59) at various combination ratios [76]. Additionally, rutin has also been relatively extensively studied for its antibacterial activity. Wang et al. (2021) found that rutin had strong antibacterial effects against planktonic *K. pneumoniae* cells and significantly reduced the production of *K. pneumoniae* biofilm [77]. A similar observation was noted when rutin was tested against both *Pseudomonas aeruginosa* and MRSA, with rutin completely inhibiting biofilm inhibition of these two bacteria [78]. The presence of rutin as well as gallic acid, ellagic acid, and corilagin in the *T. citrina* extracts suggests that the extracts may also inhibit antibiofilm formation, although further studies are needed to confirm this potential.

Luteolin is a flavonoid with a structural similarity to quercetin, but it lacks the hydroxyl group at position 3 in the C ring. Notably, both luteolin and its derivative, orientin (luteolin-8-C-glucoside), were identified in both extracts of *T. citrina* in our study. Additionally, the derivative orientin 2″-*O*-gallate was identified in the aqueous extract of *T. citrina* but was absent in the methanolic extract. Minor structural differences can significantly affect the bioactivity of flavonoids [79,80]. For example, quercetin has an MIC of 35.76 µg/mL against *E. coli*, whereas luteolin has a higher MIC of 67.25 µg/mL [79]. This suggests that the hydroxy group located at position 3 of the C ring is crucial for the antibacterial activity of flavonoids [79]. Another study reported that luteolin also inhibits the growth of *Trueperella pyogenes*, with an MIC of 78 µg/mL [81]. Transmission electron microscopy images indicated that the *T. pyogenes* cell wall and membrane were damaged following luteolin exposure. Additionally, orientin, which has been identified in several *Terminalia* species [40,47,71,82], has been widely studied for its antibacterial, antiviral, anti-aging, and anti-inflammatory properties [83,84]. Wang et al. (2021) found that orientin reduced the virulence of *S. aureus* in vivo and protected mice from *S. aureus* (70). Notably, orientin may contribute to the strong antibacterial activity reported for the *T. citrina* extracts against MRSA in our study, as the MICs for MRSA are similar to those of the non-resistant *S. aureus* strain.

Two further flavonoids (astragalin and pinocembrin) were also identified in the aqueous and methanol of *T. citrina*, respectively. Astragalin, also known as kaempferol-3-glucoside, is a naturally occurring flavonoid recognized for its various bioactivities, including anti-inflammatory, antidiabetic, antioxidant, anticancer, anti-obesity, and antifungal properties [85,86,87]. While no antibacterial activity has been reported for this compound, Ivanov et al. (2020) reported that astragalin affects the cell membrane integrity of *Candida albicans* by decreasing efflux pump expression [88]. As a result, this leads to an increased concentration of antifungal drugs within the fungal cells. The authors concluded that astragalin exhibits significant antimicrobial and antibiofilm effects on *C. albicans* [88]. Although astragalin may exert similar antibiofilm effects on bacterial strains, this remains to be tested. In contrast, the antibacterial activity of pinocembrin has been relatively well studied [89]. A propolis extract rich in pinocembrin was reported to be effective against several MRSA clinical isolates, with MICs ranging from 0.39 to 0.78 mg/mL [90]. Additionally, synergistic effects were observed when this extract was combined with cefoxitin, erythromycin, and penicillin against MRSA [90]. Another two studies reported that a similar extract inhibited the growth of *Streptococcus mutans* [91], *S. aureus*, *E. coli*, and *Candida albicans* [92]. Although the exact mechanism of action remains unclear, the pinocembrin found in *T. citrina* methanol extracts may interact synergistically with other polyphenols and conventional antibiotics, enhancing the antibacterial effect of the extract. However, this remains to be tested.

## 4. Materials and Methods

### 4.1. Materials

Mueller–Hinton broth and agar powders (Oxoid Ltd., Thebarton, Australia) were prepared according to the manufacturer’s instructions and used as bacterial growth media. All antibiotics and chemicals were purchased from Sigma-Aldrich (Castle Hill, Australia) unless otherwise specified. The extraction solvents used in this study were all analytical reagent (AR) grade and were purchased from Ajax Fine-Chemicals Ltd. (Taren Point, Australia).

### 4.2. Plant Collection and Extraction

The leaves of *Terminalia citrina* (Gaertn.) Roxb. ex Flem. were collected from East Malaysia by Dr. Stephen Teo Ping from the Forest Department of Sarawak, Malaysia. The collection was conducted under permit SFC.810-4/6/1(2021)-045. The samples were gathered at the beginning of the wet season, specifically from November to December 2021. The leaves were dried in the sun, finely ground into a powder, and packed into sealed air-tight bags for transport. The samples were stored in desiccant silica gel in a ratio of 10 parts silica to 1 part plant material to keep the samples dry at all times. The dried and ground leaf materials were thoroughly inspected for microbial contamination before extraction. A voucher specimen (Tci_Kim_2021) is stored in the School of Environment and Sciences, Griffith University, Australia. A total of 1 g of dried leaf material was weighed into a Falcon tube, and 50 mL of either methanol (Ajax Australia, AR grade) or sterile deionized water was added separately to each tube. The leaf materials were individually extracted in each solvent for 24 h at 4 °C with gentle shaking. The extracts were then filtered through the Whatman No. 54 filter paper to remove the particulates. The methanol extract was air-dried until a consistent mass was achieved upon repeated measurements. The aqueous extract was lyophilised in a freeze-drier (Dynavac FD-1C-80). The mass of the dried extracts was weighed to determine the extraction yields before being suspended in 1% DMSO (in sterile deionized water). The extracts were then filtered using a syringe-driven filter (0.22 µm, Millipore, Bayswater, Australia Ltd.) and stored at 4 °C until use.

### 4.3. Bacterial Cultures

Reference strains of *Staphylococcus aureus* (ATCC 25923), methicillin-resistant *S. aureus* (ATCC 43300), *Bacillus cereus* (ATCC 11778), and *Shigella flexneri* (ATCC 12022) were purchased from the American Type Culture Collection (ATCC). Individual strains of bacteria were streaked onto Muller–Hinton (MH) agar plates and incubated at 37 °C (or MH agar plates with 2% sodium chloride at 35 °C for both *S. aureus* and MRSA) for 24 h to obtain pure cultures. A single bacterial colony was then transferred into freshly prepared MH broth (with 2% sodium chloride for *S. aureus* and MRSA). The bacterial broth cultures were incubated at 37 °C (or 35 °C for both *S. aureus* and MRSA) until the bacteria reached the log growth phase (as determined by absorbance at 600nm). To ensure the purity of each bacterial culture, they were individually re-streaked on MH agar plates.

### 4.4. Controls for Evaluation of Antibacterial Activity

The antibacterial activities of the *T. citrina* extracts were determined using a standardized Kirby–Bauer disc diffusion and liquid microdilution assay [27]. For the disc diffusion assay, cefotaxime (30 µg), ciprofloxacin (10 µg), tetracycline (10 µg), and tetracycline discs (5 µg) (Oxoid Ltd., Thebarton, Australia) were included as positive controls. Blank discs infused with 10 µL of deionised water containing 1% DMSO were used as a negative control. The same antibiotics were used in the liquid microdilution assay, and a 1% DMSO solution was used as a negative control in that assay.

### 4.5. Disc Diffusion Assay

The antibacterial susceptibility of extracts was evaluated by measuring the zone of inhibition (ZOI) on the agar plate, as previously described [93]. To achieve consistency of the collected data, a cell count of approximately 10^8^ cells/mL (0.5 McFarland turbidity) was used for each assay. The 10^8^ cells/mL cell count was confirmed by comparing the cell suspension with 0.5 McFarland turbidity standard (Thermo Scientific, Melbourne, Australia). Plant extracts were introduced onto the blank disc at 100 µg (10 µL of the 10,000 µg/mL extracts). Then, the plates inoculated with *S. aureus* and MRSA lawns were incubated at 35 °C or at 37 °C for *B. cereus* and *S. flexneri*. All plates were incubated for 24 h before the ZOI was measured to the closest whole mm. The results are expressed as the mean of three independent experiments, each with internal triplicates (*n* = 9).

### 4.6. Liquid Microdilution Assay

Liquid microdilution assays were used to quantify the antibacterial susceptibility by determining the minimum inhibitory concentration (MIC) of each extract against the tested bacterial strains, as previously described [94,95]. Plant extracts were introduced into the wells of the first row at 10,000 µg/mL. Conventional antibiotics were assessed in parallel on all plates as positive controls at a starting concentration of 40 µg/mL. The serial doubling dilution method was employed down all columns of the plates. Plates that contained *S. aureus* and MRSA were incubated at 35 °C, whilst plates screening against *B. cereus* and *S. flexneri* were incubated at 37 °C. All cultures were incubated for 24 h before determining the MIC using *ρ*-iodonitrotetrazolium (INT) colorimetric assays. INT solution turns red in the presence of microbial growth. The lowest concentration of the extract or antibiotic that did not change colour was taken as the MIC. Any extract with an MIC of less than 1000 µg/mL is considered to exhibit noteworthy antibacterial activity, as recommended by Rios and Recio (2005) and Kuete (2010) [96,97]. The MICs were examined in duplicates in three independent experiments (*n* = 6).

### 4.7. Fractional Inhibitory Concentration (FIC) Determination

The interaction between the plant extracts and selected conventional antibiotics was also evaluated. The combinational effects were initially examined at a 1:1 ratio and analysed by calculating the sum of fractional inhibitory concentrations (∑FIC):FIC (extract) = (MIC of extract in combination with antibiotic)/(MIC of extract alone)FIC (antibiotic) = (MIC of antibiotic in combination with extract)/(MIC of antibiotic alone)∑FIC = FIC (extract) + FIC (antibiotic)

Interactions with ∑FIC values ≤0.5 were classified as synergistic; >0.5–≤1.0 as additive; >1.0–≤4.0 as indifferent; and >4.0 as antagonistic.

Any combinations that produced synergistic or additive interactions were subsequently examined across nine different ratios ranging from 10:90 to 90:10 (antibiotic:extract), with 10% increasing increments. All tests were evaluated in three independent experiments, each with internal duplicates (*n* = 6). Isobolograms were plotted using the FIC of each ratio and used to determine the synergistic ratio(s) of extract and antibiotic. Values on or below the 0.5:0.5 line indicates synergy; data points above the 0.5:0.5 line and up to and including the 1.0:1.0 line demonstrate an additive interaction; those above the 1.0:1.0 line are categorised as indifferent interaction.

### 4.8. Non-Targeted Hheadspace LC-MS Analysis

Polyphenol identification of the extract was achieved by using non-targeted headspace liquid chromatography–mass spectrometry (LC-MS) [29]. This qualitative and quantitative analysis was conducted using a Vanquish ultra-high-performance liquid chromatography (UPLC) system coupled to an Orbitrap Exploris 120 mass spectrometer (Orbitrap MS, Thermo Fisher Scientific) equipped with an electrospray ionization (ESI) source. The chromatographic separation was achieved using a Thermo Fisher Scientific Accucore RP-MS HPLC C-18 Column (2.1 mm × 100 mm, 2.6 µm) and further mass separation by Orbitrap MS. Mobile phases (A) 0.1% *v*/*v* formic acid in ultrapure water and (B) 0.1% *v*/*v* formic acid in acetonitrile (MeCN) were used in this UHPLC system, with a quaternary pump system at a flow rate of 0.6 mL/mins, and an injection volume was 2.5 µL. The elution conditions were as follows: 0–10 min, 5% B; 10–20 min, 30% B; 20–25 min, 70% B; 26–30 min, 95% B; and 31–32 min, 5% B.

The ESI source was operated in both positive and negative ion modes, with the following parameters: the capillary voltage for both modes was 3.5 kV and a 350 °C capillary temperature. The sheath gas, auxiliary gas, sweep gas, and RF lens level flow rates were set to 60, 15, 2 (arbitrary units), and 70%, respectively. The full MS scan mode was set to 85–950 *m*/*z* at a resolution of 60,000 FWHM. The data collected were analysed using Compound Discoverer™ 3.3 software, following methodology previously established by our group [27]. Databases that are used for putative compound identification included ChemSpider, mzVault, mzCloud, and the local database of Compound Discoverer™ 3.3 software (Thermo Fisher Scientific, Scoresby, Australia), as well as via comparisons with published data.

### 4.9. Toxicity Studies Using Brine Shrimp

*Artemia franciscana* nauplii lethality assay was used an initial evaluation of the toxicity of the extracts. Extract dilutions that produce ≥ 50% mortality are considered to be toxic at those concentrations. When an extract was deemed to be toxic, the extract was evaluated across a range of dilutions, and the concentration that causes 50% mortality (LC_50_) was calculated by linear regression. Approximately 50 *A. fransciscana* nauplii (Aquabuy, Silverwater, Australia) in 400 µL of artificial saltwater (32 g/L, Red Sea Pharm Ltd., Pituach, Israel) and 400 µL of individual plant extracts were added into individual wells of a 48-well plate. The same volume of potassium dichromate (2 mg/mL, AR grade, Chem-Supply, Gillman, Australia) or artificial sea water was added to separate wells as positive (1 mg/mL in the assay) and negative controls, respectively. The plates were incubated at room temperature for 24 h, and the number of dead nauplii in each well was determined to calculate the concentration that induced 50% mortality (LC_50_). The results are expressed as the mean of three independent experiments, each with triplicates (*n* = 9).

### 4.10. Toxicity Studies Using Human Dermal Fibroblasts

Human primary dermal fibroblasts (HDFs; ATCC PCS-201-012) were purchased from the American Type Culture Collection (ATCC), USA, and were used to evaluate the toxicity of the extracts. Selected conventional antibiotics (cefotaxime and tetracycline) were included for all assays as positive controls. Aliquots of the HDFs (70 µL) containing approximately 5000 cells were used in all tests. Individual 30 µL volumes of the extracts (150 µg/mL) and 30 µL of antibiotics (6 µg/mL) were added into individual wells containing 70 µL of cells, and the plates were incubated at 37 °C with 5% CO_2_ for 24 h in a humidified atmosphere. To eliminate interference from the sample colour, control wells were created with 30 µL of extract or antibiotic combined with 70 µL of cell-free media and subtracted from the test values. Wells contained cells with media only, and media only (without cells) were used as negative and blank controls, respectively. After 24 h of incubation, 20 µL of MTS reagent (Abcam, Victoria, Australia) was added into all wells, followed by incubation at 37 °C for a further 4 h. Absorbances were recorded at 490 nm using a CLARIOstar plus multi-plate reader (BMG Labtech, Victoria, Australia). All tests were repeated three times, each with internal triplicates (*n* = 9). The % cellular viability of each replicate was calculated using the following equation:$$Percentage\;of\;cellular\;viability=\frac{A_{treatment}-A_{treatment\;control}}{A_{positive\;control}-A_{negative\;control}}\times 100\%$$
where *A* = Absorbance at 490 nm. Less than 50% cellular viability indicates toxicity.

### 4.11. Statistical Analysis

Samples were tested using disc diffusion assays, and statistical analyses were conducted. The data are presented as the mean ± SEM from three independent experiments, each with internal triplicates (*n* = 9). The differences between the control and treatment groups were evaluated using one-way analysis of variance (ANOVA), with a *p*-value of <0.05 deemed statistically significant. Furthermore, all broth microdilution assays were conducted thrice for each bacterial strain on separate days, with two technical replicates per assay (*n* = 6), ensuring the reliability and reproducibility of the MIC determinations.

## 5. Conclusions

The rise in foodborne disease cases and antibiotic-resistant bacteria has increased the demand for new antibacterial treatments, especially natural therapies that use natural products. In this study, we reported a metabolomic profile of *Terminalia citrina* (Gaertn.) Roxb. ex Fleming for the first time. The aqueous and methanol extracts of *T. citrina* are rich in catechin and its derivatives, as well as vitexin 2″-*O*-*p*-coumarate, orientin, quercetin derivatives, and ellagic acid. These compounds are known for their good antibacterial bioactivities, indicating that extracts containing them may have promising antibacterial properties. The antibacterial studies reported herein demonstrated substantial antibacterial effects of the extracts against Gram-positive bacterial strains, including MRSA. This finding suggests that the antibacterial mechanisms of these extracts are more effective against the thick peptidoglycan layer found in Gram-positive bacteria than against the lipopolysaccharide outer membrane of Gram-negative bacteria. However, the antibacterial mechanism needs to be investigated further. The combination studies revealed twelve synergistic combinations. Whilst the exact mechanism behind this potentiation is still unknown, it is possible that the active phytochemicals inhibit the resistance mechanisms of tetracycline and ciprofloxacin. This inhibition could increase the likelihood of these antibiotics penetrating bacterial cells, thereby enhancing their antibacterial effects. Toxicity tests indicated that both extracts are non-toxic. Based on these findings, we suggest that *T. citrina* leaf extracts may be a promising source for developing novel antibacterial drugs, although further research is necessary. Future studies should aim to identify the minimum bacterial concentration needed to assess the properties of the antibacterial effect. Moreover, isolating compounds from the extracts and examining their synergistic effects when combined with conventional antibiotics against foodborne pathogens is crucial for understanding the mechanisms of action of these compounds. Future research should include principal component analysis (PCA) to highlight the structural similarities among different polyphenols and to predict their potential bioactivities.

## Figures and Tables

**Figure 1 antibiotics-14-00593-f001:**
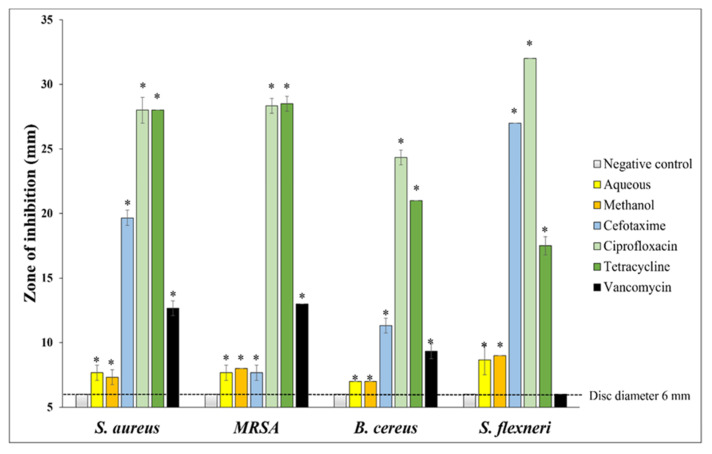
Agar disc diffusion assays using 100 µg of *T. citrina* leaf extracts (10 µL of the 10,000 µg/mL extracts) and reference antibiotics were conducted against *S. aureus*, MRSA, *B. cereus*, and *S. flexneri*. The ZOIs were measured in mm, and the disc sizes were 6 mm. Negative control discs were infused with 10 µL sterile water containing 1% DMSO. Concentrations of the cefotaxime, ciprofloxacin, tetracycline, and vancomycin controls were 30 µg, 10 µg, 10 µg, and 5 µg, respectively. Values are expressed as mean ± standard error of the mean (SEM) of three independent assays, each with internal triplicates (*n* = 9). * indicates results that are significantly different from the negative control (*p* < 0.05).

**Figure 2 antibiotics-14-00593-f002:**
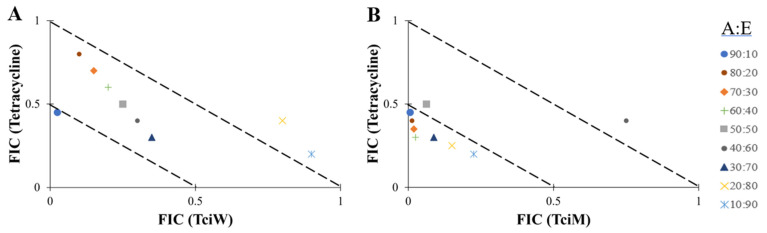
Isobolograms of tetracycline in combination with (**A**) TciW and (**B**) TciM at various combination ratios against *B. cereus*. FIC values are displayed as the means of two independent repeats *(n* = 2). Ratio = % antibiotic: % extract. FIC values below the 0.5/0.5 line represents synergy; FIC values represent additive interactions between 0.5 and 1. Only the synergistic and additive ratios are shown in these figures.

**Figure 3 antibiotics-14-00593-f003:**
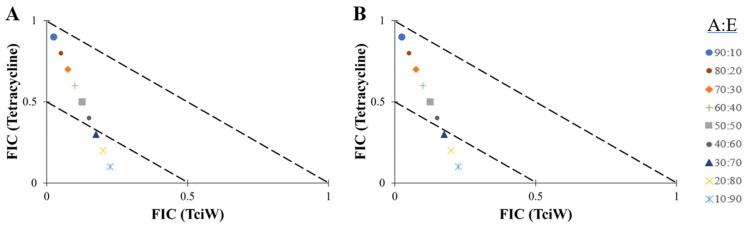
Isobolograms of TciW in combination with (**A**) ciprofloxacin and (**B**) tetracycline at various combination ratios against *S. flexneri*. FIC values are displayed as the means of two independent repeats (*n* = 2). Ratio = % antibiotic: % extract. FIC values below the 0.5/0.5 line represents synergy; FIC values represent additive interactions between 0.5 and 1. Only the synergistic and additive ratios are shown in these figures.

**Figure 4 antibiotics-14-00593-f004:**
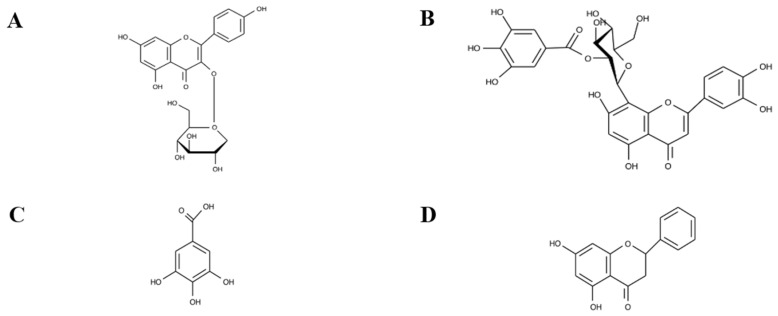
(**A**) Astragalin, (**B**) orientin 2″-O-gallate, (**C**) gallic acid, and (**D**) pinocembrin.

**Figure 5 antibiotics-14-00593-f005:**
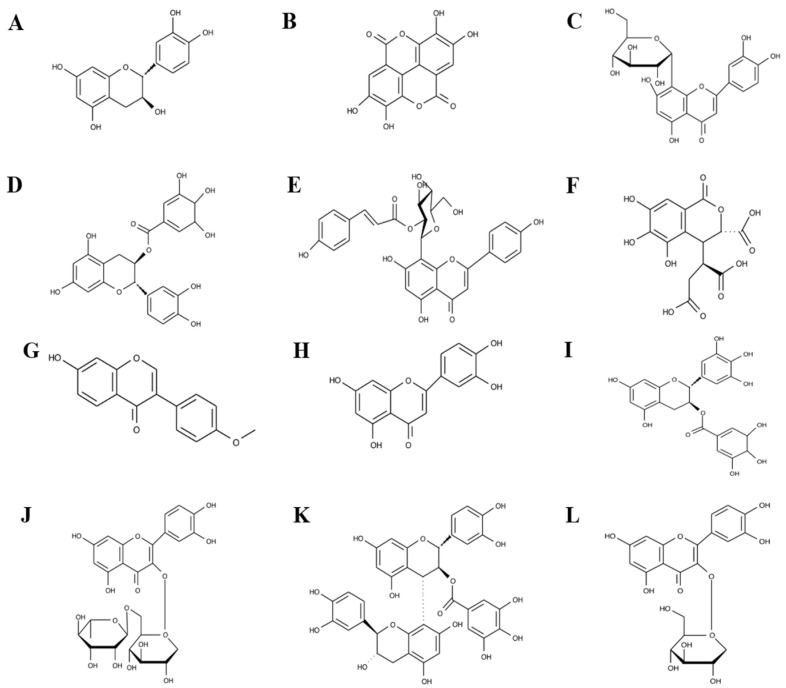
(**A**) catechin, (**B**) ellagic acid, (**C**) orientin, (**D**) (-)-epicatechin gallate, (**E**) vitexin 2″-O-p-coumarate, (**F**) chebulic acid, (**G**) formononetin, (**H**) luteolin, (**I**) (-)-epigallocatechin gallate, (**J**) rutin, (**K**) procyanidin B3 3-O-gallate, (**L**) quercetin-3β-D-glucoside, (**M**) cinnamtannin A4, (**N**) terminalin, (**O**) quercetin 3-(6″-p-hydroxybenzoylgalactoside), (**P**) 6-O-galloyl-glucose, (**Q**) 1,3,6-trigalloyl glucose, and (**R**) corilagin.

**Table 1 antibiotics-14-00593-t001:** Minimum inhibitory concentration (MIC) values for *T. citrina* extracts and the reference antibiotics against *S. aureus*, MRSA, *B. cereus*, and *S. flexneri*.

Sample	MIC (µg/mL)			
	*S. aureus*	MRSA	*B. cereus*	*S. flexneri*
Aqueous	**625**	**625**	**468.8**	2500
Methanol	**833.3**	**625**	**562.5**	2500
Cef	2.5	>10	>10	>10
Cip	1.3	3.3	0.6	2.5
Met	10	>10	>10	>10
Tet	1.9	0.5	0.3	2.5
Van	3.8	3.8	1.3	>10

Values in **bold** indicate samples showing noteworthy antibacterial activities (MIC between 400 µg/mL and 1000 µg/mL). Values in blue indicate the bacteria are susceptible to the tested antibiotic, based on CLSI M100 and M45 guidelines [25,26]. Cef = cefotaxime; Cip = ciprofloxacin; Met = methicillin; Tet = tetracycline; Van = vancomycin. The results are expressed as the mean of three independent experiments, each with internal triplicates (*n* = 9).

**Table 2 antibiotics-14-00593-t002:** ∑Fractional inhibitory concentration (∑FIC) 50:50 values between Tci extracts and the reference antibiotics against *S. aureus*, MRSA, *B. cereus*, and *S. flexneri*.

Species	Extract	∑Fractional Inhibitory Concentration (FIC)			
		Cef	Cip	Tet	Van
** *S. aureus* **	Aqueous	0.58	1.5	1.13	2.0
	Methanol	1.5	1.5	0.75	2.0
**MRSA**	Aqueous	-	0.75	1.25	2.0
	Methanol	-	1.5	0.91	2.0
** *B. cereus* **	Aqueous	-	1.5	0.75	1.0
	Methanol	-	1.25	0.56	1.0
** *S. flexneri* **	Aqueous	-	0.63	0.63	-
	Methanol	-	1.25	1.25	-

∑FIC values were evaluated in duplicate (*n* = 2). ∑FIC values >0.5–≤1.0 are classed as additive; >1.0–≤4.0 as indifferent; and >4.0 as antagonistic. - indicates not tested as one or both combination components were ineffective at all concentrations tested. Cef = cefotaxime; Cip = ciprofloxacin; Tet = tetracycline; Van = vancomycin.

**Table 3 antibiotics-14-00593-t003:** Comparison in the identified polyphenol compounds in aqueous and methanol extracts of *T. citrina* using UPLC-MS in negative-ion mode ([M-H]-1) relative to % total area.

	Rt (min)	Putative Identification	Empirical Formula	Molecular Weight	Relative Abundance (% Total Area)	
					TciW	TciM
F	2.137	(-)-Epigallocatechin gallate	C_22_ H_18_ O_11_	458.40	0.004	0.02
	2.137	Epigallocatechin 3-*O*-(4-hydroxybenzoate)	C_22_ H_18_ O_9_	426.095	0.004	0.02
	2.218	1,6-Bis-*O*-(3,4,5-Trihydroxybenzoyl) hexopyranose	C_20_ H_20_ O_14_	484.085	0.06	0.02
	2.562	Catechin	C_15_ H_14_ O_6_	290.0788	9.80	7.31
	4.072	Vitexin 2″-*O*-*p*-coumarate	C_30_ H_26_ O_12_	578.142	2.31	0.55
	5.508	3,5-Dihydroxy-2-(4-hydroxyphenyl)-4-oxo-3,4-dihydro-2H-chromen-7-yl hexopyranoside	C_21_ H_22_ O_11_	450.116	0.02	0.07
	8.043	Apigetrin	C_21_ H_20_ O_10_	432.105	0.02	0.01
	8.464	Orientin	C_21_ H_20_ O_11_	448.1	0.18	0.2
	8.845	Rutin	C_27_ H_30_ O_16_	610.153	0.10	0.06
	8.853	(-)-Epicatechin gallate	C_22_ H_18_ O_10_	442.09	0.02	0.23
	8.924	Astilbin	C_21_ H_22_ O_11_	450.116	0.01	0.01
	8.954	Quercetin-3β-D-glucoside	C_21_ H_20_ O_12_	464.095	0.08	0.09
	9.036	Orientin 2″-*O*-gallate	C_28_ H_24_ O_15_	600.112	0.04	-
	9.36	Quercetin 3-(6″-*p*-hydroxybenzoylgalactoside)	C_28_ H_24_ O_14_	584.116	0.22	0.28
	9.446	Astragalin	C_21_ H_20_ O_11_	448.101	0.07	-
	10.485	Eriodictyol	C_15_ H_12_ O_6_	288.063	0.02	0.04
	10.627	Luteolin	C_15_ H_10_ O_6_	286.048	0.02	0.05
	11.938	Apigenin	C_15_ H_10_ O_5_	270.053	0.006	0.02
	12.222	Formononetin	C_16_ H_12_ O_4_	268.073	0.02	0.21
	12.392	Pinocembrin	C_15_ H_12_ O_4_	256.074	-	0.01
	18.031	Strobopinin	C_16_ H_14_ O_4_	270.089	0.003	0.1
T	0.415	Chebulic acid	C_14_ H_12_ O_11_	356.038	0.03	0.03
	0.46	6-*O*-Galloyl-glucose	C_13_ H_16_ O_10_	332.074	0.02	0.16
	0.568	Gallic acid	C_7_ H_6_ O_5_	170.021	-	0.33
	8.13	Robinetinidol-(4-α-8)-catechin-(6,4 α)-robinetinidol	C_45_ H_38_ O_18_	866.206	0.10	0.18
	7.277	Acertannin	C_20_ H_20_ O_13_	468.09	0.06	0.05
	7.511	Ampelopsin 3′-glucoside	C_21_ H_22_ O_13_	482.106	0.04	0.06
	0.726	Corilagin	C_27_ H_22_ O_18_	634.081	0.04	0.14
	7.605	Cinnamtannin A3	C_75_ H_62_ O_30_	1442.33	0.03	0.03
	7.95	Cinnamtannin A4	C_90_ H_74_ O_36_	1730.39	0.18	0.08
	8.027	1,3,6-Trigalloyl glucose	C_27_ H_24_ O_18_	636.096	0.02	0.13
	8.153	Terminalin	C_28_ H_10_ O_16_	601.997	0.18	0.004
	8.276	Procyanidin B3 3-*O*-gallate	C_37_ H_30_ O_16_	730.153	0.08	0.19
	8.575	Ellagic acid	C_14_ H_6_ O_8_	302.006	1.00	0.98
L	8.843	Lariciresinol 4-*O*-glucoside	C_26_ H_34_ O_11_	522.20976	0.04	0.02

Flavonoids, tannins, and lignan compounds are denoted as F, T, and L, respectively.

## Data Availability

Data are either presented within the manuscript or are available from the corresponding author upon reasonable request.

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
