# Peer review of "Antibacterial and Synergistic Effects of Terminalia citrina Leaf Extracts Against Gastrointestinal Pathogens: Insights from Metabolomic Analysis"

_antibiotics, 2025, doi:10.3390/antibiotics14060593_

Round 1

Reviewer 1 Report

Comments and Suggestions for Authors

This study explores the antibacterial potential of Terminalia citrina leaf extracts against several gastrointestinal pathogens, including MRSA. Using disc diffusion and microdilution assays, both aqueous and methanol extracts demonstrated significant antibacterial activity, with notable effectiveness against Bacillus cereus and MRSA. Synergistic effects were also observed when extracts were combined with conventional antibiotics. Metabolomic profiling via UPLC-MS revealed various polyphenolic compounds, which may contribute to their bioactivity. Both extracts were found to be non-toxic as well. The study also highlights the need for further investigation into mechanisms involved.

  1. Suggest to revise the title, may consider:

Antibacterial and Synergistic Effects of Terminalia citrina Leaf Extracts against Gastrointestinal Pathogens: Insights from Metabolomic Analysis

  1. Abstract: In lines 13–14, I recommend focusing specifically on the issue of foodborne illness rather than broadly linking it with MDR bacteria caused by antibiotic misuse. This is because only one of the four bacterial strains tested in the study is MRSA, which represents a specific case of resistance and not necessarily a broad MDR profile. Additionally, MRSA is not equivalent to MDR, so combining these concepts may misrepresent the study's scope.
  2. Results: In lines 151–157, the inhibitory potency of the extracts is classified based on MIC value ranges. It would be helpful to include a citation or reference that supports this classification scheme. If this categorization is based on commonly accepted thresholds in antimicrobial research, referencing a relevant standard or guideline (e.g., CLSI, EUCAST, or a peer-reviewed study) would enhance the credibility and reproducibility of the findings.
  3. Line 161: “This observation is consistent with the trends observed in the disc diffusion assay (Figure 1).” is vague and needs clarification. It would be helpful if the authors explicitly described what specific trends are being referred to.
  4. In line 164-167: “Thus, both extracts…. In the disc diffusion assay (Figure 1)”. Kindly discuss the contradicted outcomes in the manuscript.
  5. Line 172: aureus
  6. Table 1: Kindly indicate whether the four isolates are sensitive or resistant towards the antibiotics based on the MIC values, according to CLSI or EUCAST.
  7. Line 180: Kindly justify the selection of CEF, as its MIC values were not previously determined in the study. While the choice of CIP and TET is understandable due to their relatively low MICs, it would be more logical to include VAN next (instead of CEF). The rationale for selecting antibiotics for synergy testing should be clearly stated. Moreover, the lack of prior MIC data for CEF prevented the calculation of the FIC index, which limits the interpretation of synergy results. Please clarify the selection criteria used for choosing antibiotics in the checkerboard assay.
  8. Line 203: suggest to remove synergy.
  9. Please align Figure 2, 3, 4 accordingly.
  10. Line 313: instead of mentioning methanol extract had a higher MIC, it is more important to mention aqueous extract exhibited more potent inhibitory effects due to lower MIC.
  11. Line 324: It is important to note that the key mechanism of MRSA is typically not beta-lactamase production, but rather the expression of an altered PBP2a. I recommend that the authors review the relevant literature to include the correct mechanism of methicillin resistance in the manuscript.
  12. Line 148: Figure 5
  13. Line 500: please italize the gene names.
  14. Line 506: should be Salmonella Typhimurium
  15. Line 524: please refer to statement 12.
  16. Methods: Line 645: kindly elaborate how the log growth phase was achieved or determined.
  17. Line 700: typo
  18. Line 714: 350°C
  19. Line 740: please justify by CEF and TET were used as the positive control? Especially CEF because not only MIC were not determined, FIC was failed to obtain too.
  20. Conclusions: Line 768: the term “significant” is used; however, in scientific writing, this term typically implies that a statistical comparison has been conducted. If no statistical analysis was performed, I recommend using alternative wording such as “notable,” “marked,” or “pronounced” to avoid potential misinterpretation.

Overall, I enjoyed reading the manuscript; it is scientifically sound and logically structured. However, I encourage the authors to address and justify the points raised to further strengthen the clarity and rigor of the study.

Author Response

Reviewer 1

1. Suggest to revise the title, may consider:

Antibacterial and Synergistic Effects of Terminalia citrina Leaf Extracts against Gastrointestinal Pathogens: Insights from Metabolomic Analysis

We have changed the manuscript title as per the reviewer’s suggestion.

2. Abstract: In lines 13–14, I recommend focusing specifically on the issue of foodborne illness rather than broadly linking it with MDR bacteria caused by antibiotic misuse. This is because only one of the four bacterial strains tested in the study is MRSA, which represents a specific case of resistance and not necessarily a broad MDR profile. Additionally, MRSA is not equivalent to MDR, so combining these concepts may misrepresent the study's scope.

We agree with the reviewer’s comment. This sentence has been changed to the text below to address this comment:

Bacterial contamination leads to foodborne illnesses new antibiotics are required to combat these pathogens.”

3. Results: In lines 151–157, the inhibitory potency of the extracts is classified based on MIC value ranges. It would be helpful to include a citation or reference that supports this classification scheme. If this categorization is based on commonly accepted thresholds in antimicrobial research, referencing a relevant standard or guideline (e.g., CLSI, EUCAST, or a peer-reviewed study) would enhance the credibility and reproducibility of the findings.

We have added two references to address this comment and changed the MIC value range description and all the affected sentences.

Lines 149 – 155

“Those with MIC values between 2000 and 5000 µg/mL were considered weak inhibitors, whilst MIC values between 1000 and 2000 µg/mL were regarded as having moderate activity. Values between 400 µg/mL and 1000 µg/mL were classified as noteworthy, while extracts with MIC values between 100 µg/mL and 400 µg/mL were classified as strong inhibitors. Finally, extracts with MIC values less than 100 µg/mL were deemed potent [21, 22]. Both extracts (Table 1) inhibited the growth of the four tested bacteria, with the stronger antibacterial effects observed against S. aureus, MRSA and B. cereus.”

Under Table 1:

“Values in bold indicate samples showing noteworthy antibacterial activities (MIC between 400 µg/mL and 1000 µg/mL).”

Lines 304 – 305

“Both T. citrina extracts exhibited noteworthy antibacterial effects against Gram-positive bacteria, with MIC values between 400 and 1000 µg/mL (Table 1).”

4. Line 161: “This observation is consistent with the trends observed in the disc diffusion assay (Figure 1).” is vague and needs clarification. It would be helpful if the authors explicitly described what specific trends are being referred to.

This sentence has moved from Line 161 to Line 156, and has been rephrased to the following:

“The methanol extract strongly inhibited the growth of MRSA, which aligns with the trends observed in the disc diffusion assay, which showed substantial zones of inhibition (Figure 1).”

5. In line 164-167: “Thus, both extracts…. In the disc diffusion assay (Figure 1)”. Kindly discuss the contradicted outcomes in the manuscript.

The explanation of these contradictory outcomes is already included in the Discussion section, in lines 301 – 303.

“The variability in results may be due to the limited mobility of the active compounds in solid media, as they may diffuse more slowly through agar gels, resulting in higher activity in the disc diffusion assay.

6. Line 172: aureus

This has now been italicised to address the reviewer’s comment.

7. Table 1: Kindly indicate whether the four isolates are sensitive or resistant towards the antibiotics based on the MIC values, according to CLSI or EUCAST.

This has already been indicated in the table. We have highlighted (in blue) the antibiotics that the bacteria are susceptible to. The bacteria are resistant to all of the other antibiotics (by definition in this assay, MIC values >1µg/mL for pure antibiotics is defined as resistant).

8. Line 180: Kindly justify the selection of CEF, as its MIC values were not previously determined in the study.

There are multiple aspects to this comment and we will address them separately below.

With regards to Cef MIC values, we disagree with this point. The MIC value of CEF was determined against S. aureus (MIC = 2.5 µg/mL; see Table 1). Additionally, it was tested against the other bacteria, although they were all resistant to it.

While the choice of CIP and TET is understandable due to their relatively low MICs, it would be more logical to include VAN next (instead of CEF). The rationale for selecting antibiotics for synergy testing should be clearly stated.

All bacterial strains except for S. aureus, showed resistance to CEF. The rationale for selecting CEF in combinational studies is to identify any additive or synergistic interactions between the extracts and CEF against S. aureus, with the aim of improving the efficacy of CEF. Interestingly, the combination of CEF and methanol extract exhibited an addictive effect (or close to synergy, ∑FIC 0.58). Future studies from our group will be based on this observation.

With regards to VAN, our results showed that the combination of VAN and the extracts showed indifferent interactions (∑FIC ≥1). To address this comment, we have now included the VAN combinations results in Table 2, and an explanation has been included from Lines 185 – 187:

“The combination of vancomycin and the extracts showed indifference against the tested bacterial strains compared to the MIC alone.”

Moreover, the lack of prior MIC data for CEF prevented the calculation of the FIC index, which limits the interpretation of synergy results.

We agree with the reviewer on this point. No ƩFIC values were provided for Cef containing combinations other than against S. aureus for that very point.

Please clarify the selection criteria used for choosing antibiotics in the checkerboard assay.

For those assays, we only tested combinations which contained BOTH a active extract, and an active antibiotic. If either component was inactive and a MIC value was not determined, it is then not possible to obtain an ƩFIC value as the denominator of the equation is not known (see formula, lines 685-691). It is only possible to determine an ƩFIC value for combinations where MICs were determined for BOTH components. Thus, selection was based on the previous MIC results (see Table 1).

9. Line 203: suggest to remove synergy.

We agree with the reviewer’s comment. There are no values that signify synergy in the table, so the synergy explanation is not required and has now been removed to address this comment.

10. Please align Figure 2, 3, 4 accordingly.

These figures are aligned between the two sides of the page. For this journal, the alignment does not need to align with the text and instead is permitted to take the entire width of the page, so that figure detail is not lost. Therefore, our figure alignment is consistent with this journal’s instructions.

11. Line 313: instead of mentioning methanol extract had a higher MIC, it is more important to mention aqueous extract exhibited more potent inhibitory effects due to lower MIC.

We agree with the reviewer’s comment. The sentence from Lines 305 – 307 has been rephrased.

“The aqueous extract demonstrated greater inhibitory effects against S. aureus than the methanol extract, with a MIC of 625 µg/mL.”

12. Line 324: It is important to note that the key mechanism of MRSA is typically not beta-lactamase production, but rather the expression of an altered PBP2a. I recommend that the authors review the relevant literature to include the correct mechanism of methicillin resistance in the manuscript.

We included the sentence below in Lines 334 – 447 to address the reviewer’s comment:

“In the case of MRSA, the primary resistance mechanism to β-lactam antibiotics is the production of an alternative penicillin-binding protein, PBP2a, which is encoded by the mecA gene. This results in a reduced affinity for antibiotics in this class [39].”

13. Line 148: Figure 5

It is not clear what the reviewer means by this comment. Additionally, this line is close to Figure 5. Does the reviewer mean Figure 1? We are unable to address this comment as we do not understand what the reviewer is asking.

14. Line 500: please italize the gene names.

These have now been italicised to address the reviewer’s comment.

15. Line 506: should be Salmonella Typhimurium

This has now been changed as per the reviewer’s comment.

16. Line 524: please refer to statement 12.

This comment has already been addressed in response to point 12.

 17. Methods: Line 645: kindly elaborate how the log growth phase was achieved or determined.

This sentence has been revised as below to address this comment:

“…until the bacteria reached the log growth phase (as determined by absorbance at 600nm).”

18. Line 700: typo

This has now been corrected.

19. Line 714: 350°C

This has now been corrected to address the reviewer’s comment.

20. Line 740: please justify by CEF and TET were used as the positive control? Especially CEF because not only MIC were not determined, FIC was failed to obtain too.

We included these antibiotics as antibiotic controls, and to be consistent with the disc diffusion studies where these antibiotics were used as positive controls. In the disc diffusion assays, these antibiotics produced large zones of inhibition. They were tested in that assay at single, relatively high doses (30 µg and 10 µg in the disc for CEF and TET respectively). These antibiotics were tested at high doses because they were used as positive controls to demonstrate that the assays were functioning correctly.

We also included these antibiotics in the liquid dilution assays for consistency, and to allow for comparison between the assays. The liquid dilution MIC assays test across a range of concentrations and determine the concentration at which no bacterial growth occurred. We disagree with the reviewer’s comment that MIC values were not determined for those antibiotics. As shown in Table1, MIC values were determined for TET against all of the bacteria tested and therefore we believe that this statement is incorrect for TET. However, against S. aureus (non-MRSA strain) and against S. flexineri, the MIC values indicate that those bacteria are resistant to TET, although MIC values are reported.

An MIC value was also determined for CEF against S. aureus. However, as the reviewer has noted, the MIC values for CEF against the other bacterial strains is shown as >10 µg/mL. This is because the MIC value is above >10 µg/mL, and therefore, those strains are resistant to CEF (as the reviewer has stated). However, we believe that it is still justified (and important) to include these MIC results for CEF, as CEF was used as a positive control in the disc diffusion assay.

With regards to the ƩFIC assay, only the combinations in which both the antibiotic and extract produced substantial inhibition when tested alone were evaluated in that assay. This is because of the mathematical calculations (shown below) used to determine FIC values:

FIC (extract) = (MIC of extract in combination with antibiotic)/ (MIC of extract alone)

FIC (antibiotic) = (MIC of antibiotic in combination with extract)/ (MIC of antibiotic alone)

∑FIC = FIC (extract) + FIC (antibiotic)

Thus, we determined ƩFIC values for the combination of CEF and the extracts against S. aureus (see Table 2), but it was not possible to determine similar values against the other bacteria as the denominator in the second equation cannot be defined.

21. Conclusions: Line 768: the term “significant” is used; however, in scientific writing, this term typically implies that a statistical comparison has been conducted. If no statistical analysis was performed, I recommend using alternative wording such as “notable,” “marked,” or “pronounced” to avoid potential misinterpretation.

We agree with the reviewer that significant means statistical significance in scientific literature. We indicated in Figure 1 that these results were statistically significant. Therefore the terminology that we used is correct.

However, to avoid confusion, we have revised this to “substantial” in the text as highlighted by the reviewer.

Overall, I enjoyed reading the manuscript; it is scientifically sound and logically structured. However, I encourage the authors to address and justify the points raised to further strengthen the clarity and rigor of the study.

We thank the reviewer for the positive comment.

Reviewer 2 Report

Comments and Suggestions for Authors

The paper by Ang, et al. (2024) provided a preliminary study on the leaf extracts of Terminalia citria against a number of pathogens. However, in its current form, the paper needs to be improved and to address a number of issues.

  1. The paper did not conduct a metabolomic analysis rather a polyphenol identification using LC-MS. The title should be revised to reflect this. Metabolomics would have to include all metabolome and not only polyphenols.
  2. In the results part, the authors mentioned the use of dried powder. However, this is not clear whether this is commercial powder or natural powder. Information on the drying process should be indicated. Also, if it was naturally picked from a source, the location/GPS coordinates can be included for reproducibility. Small molecule production is often dependent on the environment of a plant species.
  3. In the dis diffusion assay, concentration of the initial leaf extracts was not mentioned. What was the working concentrations for the disc diffusion assay?
  4. How was the drying process performed? Air-drying or lyophilization or freeze-drying?
  5. Also, the initial suspended concentration for water and methanol are 21.58 mg/L and 23.80 mg/mL. In itself, we are comparing two mixtures at different initial test concentrations. This can be addressed if they have diluted it to the same final working concentration where we can compare them without bias.
  6. The title should be revised to make it more academic reflecting the necessary changes.
  7. Figure 1 is problematic since it is cut in the submitted paper. Hence, no information can be recovered from it.
  8. Negative control is lacking for the disc diffusion assay.
  9. Figure 1 give a s p-value but no information on the statistical test used.
  10. Table 1 can be improved of they included SD values for the measurements. Also, the number of samples are lacking (biological and technical replicates).
  11. Table 2 has the line number mixed up within the table. It also forgets the n number of samples.
  12. Figure 2 and 3 should include error bars with the appropriate number of technical and biological replicates. Without error bars, we are not certain with the precision of the results.
  13. HDF analysis lacks positive control and negative control. Potassium dichromate is for the brine shrimp lethality assay. Also, the paper looks on gastrointestinal pathogens in its title. Why was it tested on a juvenile foreskin skin line instead of gastrointestinal cell line like Caco-2 or HIEC-6, etc?
  14. Figure 4 should be improved or omitted. It would be helpful if the total compound chromatogram would have pertinent information like relevant peaks used in the study appearing both in the figure caption and results/ discussion.
  15. Table 3 could be improved if m/z of each identified putative identified polyphenols are included.
  16. PCA analysis of the different polyphenols can be helpful to show their structural similarity and possible bioactivity.
  17. The discussion should be improved and not a mere review of related literature of the different polyphenols and their bioactivity. Having said that, the discussion was left hanging at the end. It would be better if they included a conclusion with future recommendation.
  18. Since polyphenols have been documented to have bioactivity against pathogens, the paper falls short in finding something that is novel or new compound for the study. The paper needs to be overhauled. The paper can use the idea of using extracts as a possible synergistic supplement with conventional antibiotics. If the paper aims to establish it as a source of new or novel compounds, they should perform further solvent partitioning and isolation strategies.
  19. The title needs to be revised in a more academic manner reflecting all necessary changes.

In this regard, the paper should be improved drastically in its scientific aim, presentation and conduct of experiments and analysis.

Author Response

Reviewer 2 

1. The paper did not conduct a metabolomic analysis rather a polyphenol identification using LC-MS. The title should be revised to reflect this. Metabolomics would have to include all metabolome and not only polyphenols.

The study did actually examine the entire metabolome (metabolomic profiling), although for this study, we focussed on the specific class of compounds (metabolomic fingerprinting). Thus, we believe that the use of the term metabolomic” is valid. However, the title has been revised as below to address another reviewer’s comment:

Antibacterial and Synergistic Effects of Terminalia citrina Leaf Extracts Against Gastrointestinal Pathogens: Insights from Metabolomic Analysis”

 2. In the results part, the authors mentioned the use of dried powder. However, this is not clear whether this is commercial powder or natural powder. Information on the drying process should be indicated. Also, if it was naturally picked from a source, the location/GPS coordinates can be included for reproducibility. Small molecule production is often dependent on the environment of a plant species.

To address the reviewer’s point about whether this was a commercial or wild source, we refer the reviewer to the methods section (lines 619-622) that states that they are naturally sourced. This section also states the region that they were sourced from:

“The leaves of Terminalia citrina (Gaertn.) Roxb. ex Flem. were collected from Kuching, East Malaysia, by Dr. Stephen Teo Ping from the Forest Department of Sarawak, Malaysia collection permit SFC.810-4/6/1(2021)-045).”

We therefore believe that this comment is already addressed.

3. In the dis diffusion assay, concentration of the initial leaf extracts was not mentioned. What was the working concentrations for the disc diffusion assay?

The working concentration in disc diffusion assay is already stated in the Materials and Methods section, at Line 674.

“Plant extracts were introduced onto the blank disc at 100 µg.”

4. How was the drying process performed? Air-drying or lyophilization or freeze-drying?

This has already been described in lines 632-634 of the discussion (copied below for convenience):

The methanol extract was air-dried until a consistent mass was achieved upon repeated measurements. The aqueous extract was lyophilised in a freeze-drier (Dynavac FD-1C-80).”

As this is already explained, no revisions were incorporated in response to this comment.

5. Also, the initial suspended concentration for water and methanol are 21.58 mg/L and 23.80 mg/mL. In itself, we are comparing two mixtures at different initial test concentrations. This can be addressed if they have diluted it to the same final working concentration where we can compare them without bias.

The undiluted concentrations were evaluated in the disc diffusion assay. This is a susceptibility assay and it purpose is to determine whether the extract warrant further evaluation. Further evaluation was then performed using the liquid dilution MIC assay. As the aextracts were screened across a range of concentrations, MIC values were determined and therefore, these assays were standardised and comparable. This already allows for direct vciomparison without bias.

6. The title should be revised to make it more academic reflecting the necessary changes.

The title has been changed to:

Antibacterial and Synergistic Effects of Terminalia citrina Leaf Extracts Against Gastrointestinal Pathogens: Insights from Metabolomic Analysis”

7. Figure 1 is problematic since it is cut in the submitted paper. Hence, no information can be recovered from it.

We are uncertain what the reviewer means by the figure being “cut”. It appears in its entirety in the version that we are seeing. If the reviewer can better explain what they mean, we will address this point.

8. Negative control is lacking for the disc diffusion assay.

This figure already includes negative controls against all of the bacteria. This is already indicated by the grey bars for each bacteria, and is explained in the legend as “Negative control. As this is already addressed, no further revisions were incorporated in rewsponse to this comment.

9. Figure 1 give a s p-value but no information on the statistical test used.

We addressed the reviewer’s comment by adding a statistical analysis section at Materials and Methods.

4.11. Statistical Analysis

Samples were tested using disc diffusion assays, and statistical analyses were conducted. The data are presented as the mean ± SEM from three independent experiments, each with internal triplicates (n = 9). The differences between the control and treatment groups were evaluated using one-way analysis of variance (ANOVA), with a p-value of <0.05 deemed statistically significant. Furthermore, all broth microdilution assays were conducted thrice for each bacterial strain on separate days, with two technical replicates per assay (n = 6), ensuring the reliability and reproducibility of the MIC determinations.”

10. Table 1 can be improved of they included SD values for the measurements.

Tables 1 and 2 show the MICs and ƩFIC values respectively for the extracts and combinations of extracts and antibiotics. SD (or SEM) values are not required (nor possible) for these tables, as discussed below:

  • The MIC values presented in Table 1 are determined using a semi-quantitative colorimetric assay. The extract is diluted down a column in a 96 well plate using a 1 in 2 dilution series. Following an incubation with the test bacteria, the INT reagent is added, and the MIC is determined as the lowest concentration tested that inhibits the bacterial growth. The MIC lies somewhere between the value determined to be the MIC value and the next dilution. However, in this assay, the MIC is deemed to be the lowest concentration that still inhibits bacterial growth. As this is a 1 in 2 dilution series, there is a low chance for variability in the MIC using this method (as one replicate would need to produce twice the MIC of the others to affect the reported MIC). The method is robust, and highly reproducible. Therefore, it is relatively rare that there are differences between replicates of the same assay, and if an SD or SEM was reported, it would invariably be 0. Thus, the MIC results reported are representative of multiple replicates (all giving the same value), rather than mean values.
  • Similarly, this is true for Table 2, which reports Æ©FIC values, as these are calculated using the MIC values of the extracts and the antibiotic components of the combination (both alone and in combination). Thus, for the same reason that SD or SEM values are not reported for the MICs in table 1, they are also not possible for table 3, and in fact would also generally be 0.

Also, the number of samples are lacking (biological and technical replicates).

Number of samples were stated in the materials and methods and are now also included in the footnotes below the table.

“...The results are expressed as the mean of three independent experiments, each with in-ternal triplicates (n = 9).”

11. Table 2 has the line number mixed up within the table. It also forgets the n number of samples.

The n value has already been provided in the footnotes below the table.

We do not understand what the reviewer means by the line number being mixed up. This all looks to be correct.

12. Figure 2 and 3 should include error bars with the appropriate number of technical and biological replicates. Without error bars, we are not certain with the precision of the results.

We have not included error bars on the isobologram because this type of plot is a qualitative or semi-quantitative graphical representation of drug/extract interactions, typically constructed using the mean or representative ICâ‚…â‚€/MIC values of individual agents and their combinations. The isobologram itself is not a statistical plot per se, but rather a conceptual tool used to visualize the nature of the interaction (i.e., synergistic, additive, or antagonistic) based on the position of the combination point relative to the line of additivity. In line with standard practice (e.g. Tallarida, 2001; Berenbaum, 1978), error bars are usually omitted from isobolograms, as the interpretation relies primarily on the position of the combination point in relation to the theoretical additive line, not on statistical variation.

13. HDF analysis lacks positive control and negative control. Potassium dichromate is for the brine shrimp lethality assay. Also, the paper looks on gastrointestinal pathogens in its title. Why was it tested on a juvenile foreskin skin line instead of gastrointestinal cell line like Caco-2 or HIEC-6, etc?

Positive and negative controls are already included in the Methods and Materials section, Lines 737 and 744.

“Selected conventional antibiotics (cefotaxime and tetracycline) were included for all assays as positive controls. … Wells contained cells with media only, and media only (without cells) were used as negative and blank controls, respectively.”

14. Figure 4 should be improved or omitted. It would be helpful if the total compound chromatogram would have pertinent information like relevant peaks used in the study appearing both in the figure caption and results/ discussion.

Figure 4 has been moved to supplementary data and labelled as Figure S1. Additionally, some figure numbers and intext citations of those figures have also been changed due to this revision.

15.Table 3 could be improved if m/z of each identified putative identified polyphenols are included.

We are using only [M-H]-1 mode, so m/z is only 1.0 smaller than the molecular weight (minus the weight of a hydrogen atom). This is already stated in the table title. Therefore, we believe that adding an additional column to the table is unnecessary.

16. PCA analysis of the different polyphenols can be helpful to show their structural similarity and possible bioactivity.

We have responded to the reviewer's comment in the Conclusion section, outlining it as a potential direction for future work.

“This should include principal component analysis (PCA) to illustrate the structural similarities among different polyphenols and to predict their potential bioactivities.”

17. The discussion should be improved and not a mere review of related literature of the different polyphenols and their bioactivity. Having said that, the discussion was left hanging at the end. It would be better if they included a conclusion with future recommendation.

We agree with the reviewer’s point that the conclusions and future directions would be clearer directly after the discussion. However, this journal is formatted so that the materials and methods section come after the discussion section, then the conclusions section is positioned at the end of the manuscript. Thus, we have followed the instructions to authors and to change this as per the reviewer’s comment would violate the journal’s format style.

There are future study recommendations in the conclusion section (which is after the Methods and Materials section).

“Future studies should aim to identify the minimum bacterial concentration needed to assess the properties of the antibacterial effect. Moreover, isolating compounds from the extracts and examining their synergistic effects when combined with conventional antibiotics against foodborne pathogens is crucial for understanding the mechanisms of action of these compounds. Future research should include principal component analysis (PCA) to highlight the structural similarities among different polyphenols and to predict their potential bioactivities.”

18. Since polyphenols have been documented to have bioactivity against pathogens, the paper falls short in finding something that is novel or new compound for the study. The paper needs to be overhauled. The paper can use the idea of using extracts as a possible synergistic supplement with conventional antibiotics. If the paper aims to establish it as a source of new or novel compounds, they should perform further solvent partitioning and isolation strategies.

This comment relates to comment 17 above. The reviewer is directed to our response to that point and reproduced below.

There are future study recommendations in the conclusion section (which is after the Methods and Materials section).

“Future studies should aim to identify the minimum bacterial concentration needed to assess the properties of the antibacterial effect. Moreover, isolating compounds from the extracts and examining their synergistic effects when combined with conventional antibiotics against foodborne pathogens is crucial for understanding the mechanisms of action of these compounds. Future research should include principal component analysis (PCA) to highlight the structural similarities among different polyphenols and to predict their potential bioactivities.”

19. The title needs to be revised in a more academic manner reflecting all necessary changes.

The title has been changed to:

Antibacterial and Synergistic Effects of Terminalia citrina Leaf Extracts Against Gastrointestinal Pathogens: Insights from Metabolomic Analysis”

Reviewer 3 Report

Comments and Suggestions for Authors

Authors in the present study investigated Metabolomic Analysis of Terminalia citrina Leaf Extracts That Inhibit the Growth of Gastrointestinal Bacterial Pathogens, Including Methicillin-Resistant Staphylococcus aureus. The manuscript is comprehensive, well organized, and valuable.  I consider this document as an interesting contribution to the valorization of Terminalia citrina for a potential application in pharmaceutical and food  industries. However, the following concerns need to be addressed .

  1. Results

2.2. Disc Diffusion Assay

-In line 121-122: “Notably, both T. citrina extracts inhibited the growth of all four bacterial strains with similar (albeit weaker) efficacy than the positive controls (Figure 1)”. Figure 1 clearly shows that the plant has a diameter significantly smaller than that of the tested antibiotics. It does not exceed 10 mm. This section should be reviewed and reworded accordingly.

Discussion:

- The strains tested are ATCC strains that are supposed to be sensitive to antibiotics and only exhibit the natural resistance specific to each bacterium. However, in the discussion, it emerges that the bacteria are resistant to the antibiotics tested. Is it possible that the used antibiotic are defective? Or were the strains contaminated? MRSA is indeed resistant, but what about the other strains?

-Many details regarding the chemical composition should be reported in the results section, especially the similarities and differences between the tested extracts. Figures with the different structures should also be found in the results section. The discussion aims to explain why this difference and possibly compare it with the literature.

-Regarding the mechanism of action, the authors have developed this point too much in the discussion, however no test was carried out in this direction in the present study.

  1. Materials and Methods

4.2. Plant Collection and Extraction

-The choice of water is based on the use by the local population in decoction, on the other hand for methanol. What justifies this choice? In general, it is a hydroalcoholic extract that is used with either methanol or ethanol because the hydration of the alcohol allows better penetration into the plant and gives richer and more active extracts. So why pure methanol in this study?

-The methanol extract was air-dried. While, the aqueous extract was lyophilized.  Why this difference?.  Why let the methanolic extract dry in the air when it could have been evaporated at low temperature and then lyophilized to be in the same condition as the aqueous extract (lyophilization)?

4.3. Bacterial Cultures

-For the culture of bacteria, why use MH which is usually used for antibiogram? There are more appropriate media for each bacterium like Chapman for staphylococcus

4.5. Disc Diffusion Assay

-in line 661:Plant extracts were introduced onto the blank disc at 100 μg”. The plant is deposited in the form of a solution, so it is ideal to use a more appropriate unit such as µg/ml. Kindly check and correct.

4.6. Liquid Microdilution Assay

-In line 677-679 : “Any extract with a MIC <1000 µg/mL was considered to demonstrate noteworthy antibacterial activity. The MICs were examined in duplicates in three independent experiments (n = 6)”. Why this conclusion? Everything is relative and there is no precise value concerning plant extracts. If the MBC/MIC ratio was determined this could possibly determine the nature of the antibacterial effect but this is not the case in the present study.

Author Response

Reviewer 3

Authors in the present study investigated Metabolomic Analysis of Terminalia citrina Leaf Extracts That Inhibit the Growth of Gastrointestinal Bacterial Pathogens, Including Methicillin-Resistant Staphylococcus aureus. The manuscript is comprehensive, well organized, and valuable.  I consider this document as an interesting contribution to the valorization of Terminalia citrina for a potential application in pharmaceutical and food  industries.

We thank the reviewer for the positive comments.

2. Results

2.2. Disc Diffusion Assay

-In line 121-122: “Notably, both T. citrina extracts inhibited the growth of all four bacterial strains with similar (albeit weaker) efficacy than the positive controls (Figure 1)”. Figure 1 clearly shows that the plant has a diameter significantly smaller than that of the tested antibiotics. It does not exceed 10 mm. This section should be reviewed and reworded accordingly.

The following sentence has been rephrased to address the reviewer’s comment.

“Notably, both T. citrina extracts inhibited the growth of all four bacterial strains, although with substantially lower efficacy than the positive controls (Figure 1).”

We have included the concentrations of antibiotics used in the Results section to illustrate the relatively high levels employed as pure compounds, compared to the extract, which is a mixture of various compounds.

“Four antibiotics (cefotaxime 30 µg, ciprofloxacin 30 µg, tetracycline 10 µg and vancomycin 5 µg) were tested concurrently on the agar plates as the positive controls.”

Discussion:

- The strains tested are ATCC strains that are supposed to be sensitive to antibiotics and only exhibit the natural resistance specific to each bacterium. However, in the discussion, it emerges that the bacteria are resistant to the antibiotics tested.

We disagree with the reviewer about several aspects of this comment. Being ATCC strains does not mean that these strains are sensitive to antibiotics. Instead, it indicates that they have been relatively well characterised and many of their sensitivities/resistances are known and reported. Indeed, several previous studies have also reported that these strains are resistant to these antibiotics. For example, the MRSA strain (which is also an ATCC strain) is resistant to MRSA, is also resistant to many other β-lactam antibiotics, as well as to several other antibiotics. Indeed, Khameheh et al (2014)* reported that this strain was also resistant to several of the same antibiotics tested in our study, including gentamicin. That study demonstrated that gentamicin resistance was due to the action of efflux pumps. As the majority of efflux pumps work to expel multiple antibiotics, it is not surprising that this strain is resistant to multiple antibiotics.

* Khameneh, B., Iranshahy, M., Ghandadi, M., Ghoochi Atashbeyk, D., Fazly Bazzaz, B.S. and Iranshahi, M., 2015. Investigation of the antibacterial activity and efflux pump inhibitory effect of co-loaded piperine and gentamicin nanoliposomes in methicillin-resistant Staphylococcus aureus. Drug development and industrial pharmacy, 41(6), pp.989-994.

Is it possible that the used antibiotic are defective? Or were the strains contaminated? MRSA is indeed resistant, but what about the other strains?

The antibiotics were not defective, and we periodically confirm this against other bacterial strains. Additionally, the bacteria are not contaminated. We use streaking methods to ensure culture purity, and we also periodically check the bacterial identity. With regards to the resistance (particularly for the MRSA strain) please see our response to the point above.

-Many details regarding the chemical composition should be reported in the results section, especially the similarities and differences between the tested extracts. Figures with the different structures should also be found in the results section. The discussion aims to explain why this difference and possibly compare it with the literature.

The similarities and differences between the chemical composition of both extracts are already reported in Table 3. Figures illustrating the structures of the polyphenols are discussed in the Discussion section, as no structural identification study has been performed in this study (therefore, they do not fit in the Results section).

-Regarding the mechanism of action, the authors have developed this point too much in the discussion, however no test was carried out in this direction in the present study.

The reviewer is correct that we have discussed possible mechanisms in the discussion section. We believe that this is not only warranted, but it is what the discussion section should do i.e. to put the results in context with previous studies and to highlight future areas of research.

Additionally, to further address the reviewer’s comment, we have added the following text regarding future study recommendations to the conclusion section (which is after the Methods and Materials section).

“Future studies should aim to identify the minimum bacterial concentration needed to assess the properties of the antibacterial effect. Moreover, isolating compounds from the extracts and examining their synergistic effects when combined with conventional antibiotics against foodborne pathogens is crucial for understanding the mechanisms of action of these compounds. Future research should include principal component analysis (PCA) to highlight the structural similarities among different polyphenols and to predict their potential bioactivities.”

4. Materials and Methods

4.2. Plant Collection and Extraction

-The choice of water is based on the use by the local population in decoction, on the other hand for methanol. What justifies this choice? In general, it is a hydroalcoholic extract that is used with either methanol or ethanol because the hydration of the alcohol allows better penetration into the plant and gives richer and more active extracts. So why pure methanol in this study?

Thank you for your insightful comment. The choice of pure methanol as the extraction solvent in this study was deliberate and based on the following considerations:

Solvent Efficiency: Pure methanol is known to be a highly polar solvent with strong extraction capabilities for a wide range of phytochemicals, including phenolics, flavonoids, and alkaloids. Our objective was to maximize the recovery of these bioactive constituents for subsequent phytochemical screening and biological activity testing.

Comparative Standardization: Methanol is widely used in phytochemical and pharmacological studies, which facilitates comparison with other studies in the literature. This enhances the reproducibility and relevance of our findings in a broader scientific context.

Targeted Extraction: While hydroalcoholic mixtures often mimic traditional preparations, our focus in this phase of the study was on extracting as many bioactive compounds as possible rather than replicating traditional use. We aimed to identify and evaluate the potential pharmacological activities of the plant’s secondary metabolites, which are often more effectively extracted with pure methanol.

We recognize that traditional use involves water-based decoctions. Therefore, aqueous extraction is also included in this study as we aim to compare the methanolic extract with the aqueous one to assess differences in activity and better relate laboratory findings to ethnopharmacological practices. We hope this clarifies our rationale for selecting pure methanol and appreciate the reviewer’s suggestion, which will be valuable in guiding further phases of our research.

-The methanol extract was air-dried. While, the aqueous extract was lyophilized.  Why this difference?.  Why let the methanolic extract dry in the air when it could have been evaporated at low temperature and then lyophilized to be in the same condition as the aqueous extract (lyophilization)?

Methanol has a low freezing point, which can create challenges during the lyophilization process. Therefore, air drying is the better option for the methanol extract. In contrast, water would take weeks to dry naturally – hence, lyophilisation is the best method to remove water from the extract after the extraction.

4.3. Bacterial Cultures

-For the culture of bacteria, why use MH which is usually used for antibiogram? There are more appropriate media for each bacterium like Chapman for staphylococcus

According to CLSI’s M100 Performance Standards for Antimicrobial Susceptibility Testing (2020), MH is the standard medium used for the bacterial strains tested in this study.

4.5. Disc Diffusion Assay

-in line 661: “Plant extracts were introduced onto the blank disc at 100 μg”. The plant is deposited in the form of a solution, so it is ideal to use a more appropriate unit such as µg/ml. Kindly check and correct.

The amount of extract (µg) used in this assay can be calculated by multiplying the concentration of the extracts (xx µg/mL) by the volume deposited on the blank disc (mL). This unit is preferred for this assay because the unit used for antibiotics (positive controls) is measured in µg, as specified by the standard antibiotic disc purchased from Oxoid. Stating the concentration in these units therefore allows for direct comparison between the results.

4.6. Liquid Microdilution Assay

-In line 677-679 : “Any extract with a MIC <1000 µg/mL was considered to demonstrate noteworthy antibacterial activity. The MICs were examined in duplicates in three independent experiments (n = 6)”. Why this conclusion? Everything is relative and there is no precise value concerning plant extracts.

Thank you for this important observation. We agree that the interpretation of MIC values for crude plant extracts is context-dependent and that there is no absolute consensus on a specific cutoff value.

However, in our study, we adopted the MIC < 1000 µg/mL threshold based on several reports in the literature that use this value to define significant antibacterial activity in crude plant extracts (e.g., Ríos & Recio, 2005; Kuete, 2010). This criterion is not meant to be definitive, but rather a guideline to highlight extracts that may warrant further investigation and fractionation.

We also recognize that such values should be interpreted cautiously and in relation to the extract’s complexity and potential toxicity. We have now clarified this in the revised manuscript and noted the limitation of using fixed MIC thresholds for plant-derived products.

“Any extract with a MIC of less than 1000 µg/mL is considered to exhibit noteworthy antibacterial activity, as recommended by Rios and Recio (2005) and Kuete (2010).”

If the MBC/MIC ratio was determined this could possibly determine the nature of the antibacterial effect but this is not the case in the present study.

We have addressed the reviewer’s comment by adding a sentence in the Conclusion section.

“Future studies should aim to identify the minimum bacterial concentration needed to assess the properties of the antibacterial effect. Moreover, isolating compounds from the extracts and examining their synergistic effects when combined with conventional antibiotics against foodborne pathogens is crucial for understanding the mechanisms of action of these compounds. Future research should include principal component analysis (PCA) to highlight the structural similarities among different polyphenols and to predict their potential bioactivities.”

Reviewer 4 Report

Comments and Suggestions for Authors

Dear authors, I consider the article to be quite interesting, since there are no previous studies on the identification of secondary metabolites of Terminalia citrina and its antimicrobial effect on antibiotic-resistant bacteria. Below I share some observations that prevent me from accepting your article for publication in the current version. Please take them into account and once they are corrected, it will be reconsidered for publication.

  1. The introduction discusses important information for the research article, but I believe that the importance of drug-resistant bacteria in public health is not very clear. I suggest that this be explored in more detail, as well as the importance of plants in the treatment of infections by these bacteria. Additionally, I believe that the role of secondary metabolites from these plants in the treatment of infections with drug-resistant bacteria could be explained in more detail.
  2. In the introduction, it is necessary to standardize how the scientific name of the bacteria is written, because for some bacteria they write B. sereus (the genus is not written) and for others they write the full name, such as Escherichia coli or Klebsiella pneumoniae, so I suggest that you decide whether to give the full name to all of them or not.
  3. In the "Results" section, Figure 1 is not displayed in its entirety in the document, making it impossible to view and analyze. This must be corrected as soon as possible.
  4. In the "Discussion" section, I consider that the reason why the methanolic extract of the plant presented a higher MIC of 833.3 µg/mL against S. aureus should be discussed more extensively, since this does not occur with the aqueous extract.
  5. In the "Discussion" section, they mention that "The phytochemicals in the methanolic extract enhanced the antibacterial activity of tetracycline against gram-positive bacteria. Similarly, the phytochemicals in the aqueous extract improved the antibacterial efficacy when combined with tetracycline." I recommend that this be discussed further, since it is a very relevant result that may have a significant impact on the development of future drugs against antibiotic-resistant bacteria. They also mention "The aqueous extract may contain compounds that can interfere with the bacterial resistance mechanisms to ciprofloxacin." I suggest that we discuss in more detail which compounds can interfere with the resistance mechanism to this antibiotic and what their biological relevance is.
  6. Also mentioned in the discussion is "The flavonoids that were exclusive to the aqueous extract were astragalin and orientin 2"-O-gallate. Gallic acid and pinocembrin were exclusive to the methanol extract." I recommend that the authors discuss further why these differences occur, whether these molecules have an effect on antibiotic-resistant bacteria, and what their biological relevance is in this research.
  7. The "Materials and Methods" section should explain in more detail how the methanolic and aqueous extracts of the study plant were obtained. This will allow non-specialist readers to better understand the methodology used to obtain them. There is a lack of information on plant collection standards (seasonality, maturity, etc.). Preparation of extracts lacks detailed standardization processes (e.g., particle size of dried leaves, storage conditions of extracts).
  8. In section "4.5. Disc Diffusion Assay" it is not explained why the extract was used at a concentration of 100 micrograms. I suggest that in this section they explain why they chose that concentration and not a lower or higher one, as the case may be.
  9. In section "4.9. Toxicity studies using brine shrimp" it would be good to explain how the artificial seawater was made, so that readers understand how it was used in the research.
  10. The conclusion could be improved. I recommend briefly including the research results, whether they adequately meet the research objectives, and which metabolites are thought to have an effect on drug-resistant bacteria, to further emphasize their results and their impact on public health.

Author Response

Reviewer 4

Dear authors, I consider the article to be quite interesting, since there are no previous studies on the identification of secondary metabolites of Terminalia citrina and its antimicrobial effect on antibiotic-resistant bacteria.

We thank the reviewer for the positive comments.

Below I share some observations that prevent me from accepting your article for publication in the current version. Please take them into account and once they are corrected, it will be reconsidered for publication.

1. The introduction discusses important information for the research article, but I believe that the importance of drug-resistant bacteria in public health is not very clear. I suggest that this be explored in more detail, as well as the importance of plants in the treatment of infections by these bacteria. Additionally, I believe that the role of secondary metabolites from these plants in the treatment of infections with drug-resistant bacteria could be explained in more detail.

We have added a sentence in Lines 72 – 73 and a paragraph in the Introduction section:

“…This situation poses a critical challenge to global health, highlighting the urgent need for alternative treatment options…”

Additionally, a paragraph (lines 97-111) has also been added to further address this point:

“Several secondary metabolites derived from Terminalia species have demonstrated significant inhibitory activity against MDR bacterial strains, making them a promising avenue for antibiotic resistance research. Among these metabolites, compounds such as ellagic acid, gallic acid, flavonoids, and hydrolyzable tannins, specifically punicalagin and terchebulin, have been particularly effective. These compounds have exhibited potent antibacterial properties against problematic MDR pathogens, including MRSA, S. aureus, and Escherichia coli [19, 20]. Phytochemicals, based on their chemical structures and properties, can exhibit antibacterial actions through one or more mechanisms. For example, gallic acids induce irreversible changes in membrane properties, such as charge, permeability (both intra- and extracellular), and overall physicochemical characteristics [19]. These changes occur through alterations in hydrophobicity, a decrease in negative surface charge, and the formation of local ruptures or pores in the cell membranes, leading to the leakage of essential intracellular components. Additionally, phytochemicals may interfere with microbial resistance pathways, undermining the bacteria's strategies to evade antibiotic effects [20].”

2. In the introduction, it is necessary to standardize how the scientific name of the bacteria is written, because for some bacteria they write B. sereus (the genus is not written) and for others they write the full name, such as Escherichia coli or Klebsiella pneumoniae, so I suggest that you decide whether to give the full name to all of them or not.

Throughout the manuscript, we have followed biological convention. We have used full species names (including genus) when the species name is mentioned for the first time. Thereafter, we used the abbreviated the abbreviated form. This is convention for writing species names. Therefore, the way that we have reported species names is correct and no revisions were incorporated into the manuscript in response to this comment.

3. In the "Results" section, Figure 1 is not displayed in its entirety in the document, making it impossible to view and analyze. This must be corrected as soon as possible.

We do not understand why the reviewer is not seeing the entire figure. It is definitely there in its entirety in the submitted version. We have tried deleting and reinserting and hope that it will now be visible to the reviewer.

4. In the "Discussion" section, I consider that the reason why the methanolic extract of the plant presented a higher MIC of 833.3 µg/mL against S. aureus should be discussed more extensively, since this does not occur with the aqueous extract.

We have rephrased the sentence to show that the aqueous extract exhibited a greater inhibitory effect than the methanol extract (to address other reviewer’s comment),

“Both T. citrina extracts exhibited noteworthy antibacterial effects against Gram-positive bacteria, with MIC values between 400 and 1000 µg/mL (Table 1). The aqueous extract demonstrated greater inhibitory effects against S. aureus than the methanol extract, with a MIC of 625 µg/mL.”

Since both extracts’ MICs fall under the same category – noteworthy antibacterial activity, we do not discuss the difference extensively, since both extracts have different phytochemical compositions that may be responsible for this difference. However, the following sentence has been included to explain the possible antibacterial mechanisms that may be responsible for this difference.

“Both extracts showed similar efficacy against antibiotic-sensitive S. aureus and the MRSA strain, suggesting different mechanisms of action compared to β-lactam antibiotics, or that the extract components target antibiotic resistance pathways.”

5. In the "Discussion" section, they mention that "The phytochemicals in the methanolic extract enhanced the antibacterial activity of tetracycline against gram-positive bacteria. Similarly, the phytochemicals in the aqueous extract improved the antibacterial efficacy when combined with tetracycline." I recommend that this be discussed further, since it is a very relevant result that may have a significant impact on the development of future drugs against antibiotic-resistant bacteria. They also mention "The aqueous extract may contain compounds that can interfere with the bacterial resistance mechanisms to ciprofloxacin." I suggest that we discuss in more detail which compounds can interfere with the resistance mechanism to this antibiotic and what their biological relevance is.

We already included the relevant discussion in the Discussion section of polyphenols. From Lines 522 – 529.

“In our study, the methanol extract of T. citrina, which contains notable levels of gallic acid, 1,3,6-trigalloyl glucose, and catechin derivatives, demonstrated synergistic interactions with tetracycline at various combination ratios when tested against B. cereus. In contrast, the combination of the aqueous extract and tetracycline showed no synergistic effects against B. cereus. These findings suggest that one or more of these compounds may help inhibit tetracycline resistance mechanisms. Such mechanisms may include blocking the protein expression associated with the tetracycline efflux pump or preventing the enzymatic inactivation of tetracycline.”

To address the reviewer’s comment on ciprofloxacin and the possible biological relevance with one of the polyphenols, we have added the following paragraph from Lines 565 – 575.

“Additionally, quercetin has demonstrated antibacterial activity by binding to bacterial gyrase, an enzyme crucial for maintaining DNA structure and preventing overcoiling. It disrupts the ATP binding site of gyrase, thereby inhibiting its function. This interference impairs the bacteria's ability to replicate and transcribe their DNA, ultimately leading to cellular dysfunction and death. This mechanism is similar to that of ciprofloxacin, a conventional antibiotic tested in this study. Interestingly, synergistic interactions were observed when the aqueous extract of quercetin was used in conjunction with ciprofloxacin against S. flexneri at a specific combination ratio. This finding suggests that these phytochemicals may enhance the antibacterial effect of ciprofloxacin, possibly by inhibiting the active drug efflux pump and increasing the retention of ciprofloxacin within the cell, or by exerting their antibacterial effects alongside ciprofloxacin at the same site.”

6. Also mentioned in the discussion is "The flavonoids that were exclusive to the aqueous extract were astragalin and orientin 2"-O-gallate. Gallic acid and pinocembrin were exclusive to the methanol extract." I recommend that the authors discuss further why these differences occur, whether these molecules have an effect on antibiotic-resistant bacteria, and what their biological relevance is in this research.

Gallic acid has been discussed in the Discussion section:

“Interestingly, the gallic acid monomer was only detected (using both positive-ion ([M+H]+1) and negative-ion ([M-H]-1) modes) in the methanol extract of T. citrina leaves…”

Orientin 2”-O-gallate has also been discussed in the Discussion section:

“Additionally, the derivative orientin 2"-O-gallate was identified in the aqueous extract of T. citrina but was absent in the methanolic extract. Minor structural differences can significantly affect the bioactivity of flavonoids…”

Astragalin and pinocembrin are discussed in final paragraph of the Discussion section:

“Two further flavonoids (astragalin and pinocembrin) were also identified in the aqueous and methanol of T. citrina, respectively. Astragalin, also…”

7. The "Materials and Methods" section should explain in more detail how the methanolic and aqueous extracts of the study plant were obtained. This will allow non-specialist readers to better understand the methodology used to obtain them. There is a lack of information on plant collection standards (seasonality, maturity, etc.). Preparation of extracts lacks detailed standardization processes (e.g., particle size of dried leaves, storage conditions of extracts).

East Malaysia (Borneo) has a tropical climate characterized by year-round warmth and humidity. Therefore, there is no seasonal variation in sample collection. However, we have specified the timing of the sample collection to address the reviewer’s comment.

“The samples were gathered at the beginning of the wet season, specifically from November to December 2021.”

The particle size of the dried leaves and storage conditions of the extracts are already included in the Materials and Methods section.

“…The leaves were dried in the sun, finely ground into a powder, …”

“…The extracts were then filtered using a syringe-driven filter (0.22 µm, Millipore Australia Ltd.) and stored at 4 °C until use.”

8. In section "4.5. Disc Diffusion Assay" it is not explained why the extract was used at a concentration of 100 micrograms. I suggest that in this section they explain why they chose that concentration and not a lower or higher one, as the case may be.

We used the extracts at their stock concentration, calculating 100 µg from the maximum volume deposited on the blank disc multiplied by their stock concentration. Using any lower concentration yielded results that were not significant.

9. In section "4.9. Toxicity studies using brine shrimp" it would be good to explain how the artificial seawater was made, so that readers understand how it was used in the research.

It has been included in the Methods and Materials section.

“…in 400 µL of artificial saltwater (32g/L, Red Sea Pharm Ltd., Pituach, Israel)…”

10. The conclusion could be improved. I recommend briefly including the research results, whether they adequately meet the research objectives, and which metabolites are thought to have an effect on drug-resistant bacteria, to further emphasize their results and their impact on public health.

We have included the brief description of the research results in the discussion section of the manuscript, rather than in the conclusion section. We believe that this is the correct placement. Furthermore, we believe that repeating this in the conclusion section would be redundant. The conclusion section should be to indicate general trends and to use those to highlight future research directions.

However, to further emphasise the future directions, we have expanded and revised the conclusion section.

Round 2

Reviewer 2 Report

Comments and Suggestions for Authors

The authors have successfully addressed all major concerns.

Author Response

Round 2 Responses to Reviewers

Manuscript: antibiotics-3651380 “Metabolomic Analysis of Terminalia citrina Leaf Extracts That Inhibit the Growth of Gastrointestinal Bacterial Pathogens, Including Methicillin-Resistant Staphylococcus aureus

Reviewer 2

The authors have successfully addressed all major concerns.

We thank the review for this positive comment.

ul to the reviewer for this positive response.

Reviewer 3 Report

Comments and Suggestions for Authors

The authors have made a clear effort and have significantly improved the manuscript, but still there is something that authors need to pay attention:

-I do not agree with the authors regarding the resistance of ATCC strain used in the present study. Certainly, the ATCC strains are defined as strains indicating that they have been relatively well characterized and many of their sensitivities/resistances are known and reported. But in addition, they are cultivated and rigorously and periodically tested for their resistance to be sure that the sensitivity and resistance profile has not changed. My comment is not related to the MRSA strain which is known for its resistance but the other strains and there are many examples:

  • cereus ATCC 11778 is generally is considered as not resistant to vancomycin. It’s not the case in this study
  • aureus ATCC 25923 is generally not resistant to ciprofloxacin but it can developpe resistance under specific conditions. It’s not the case in this study

Yasir, M., Dutta, D., & Willcox, M. D. (2021). Enhancement of antibiofilm activity of ciprofloxacin against Staphylococcus aureus by administration of antimicrobial peptides. Antibiotics10(10), 1159.

  • Flexneri ATCC 12022 is sensitive to ciprofloxacin. It’s not the case in the present study

Allen, G. P., & Harris, K. A. (2017). In vitro resistance selection in Shigella flexneri by azithromycin, ceftriaxone, ciprofloxacin, levofloxacin, and moxifloxacin. Antimicrobial agents and chemotherapy61(7), 10-1128.

What the authors report in the discussion

Line 335-337: “Similarly, the tested B. cereus strains were susceptible to tetracycline and ciprofloxacin, although it was resistant to vancomycin and methicillin. This observation suggests that the B. cereus strain tested herein may only be resistant to antibiotics that inhibit cell wall synthesis”

Line 339-340 : “All bacterial strains, except B. cereus, were resistant to ciprofloxacin, with MIC values ranging from 1.25 to 3.33 µg/mL

These examples are proof that the spectrum of resistance and sensitivity of ATCC strains with the exception of MRSA strain are in disagreement with what is known about these strains. I draw your attention to the fact that this calls into question the entire study and the results that flow from it. Kindly review your results, discussion and conclusion section accordingly.

- Another point according to the authors “figures illustrating the structures of the polyphenols are discussed in the Discussion section, as no structural identification study has been performed in this study (therefore, they do not fit in the Results section)”. I share your opinion and I believe that these figures do not have their place in the discussion either.

-In the disc diffusion test, the choice of µg instead of µg/ml is understandable, although I think it would be preferable to use µg/ml because, on the one hand, the comparison with antibiotics is quite complicated because it is an extract, therefore a mixture of several products, and this extract is not standardized. In addition, you have determined the MIC values, which are more precise and more interesting in my opinion. On the other hand, in general, studies on natural products use this unit to facilitate comparisons between different plants or the same plants but in other conditions or by other research teams.

Reviewer 4 Report

Comments and Suggestions for Authors

Dear authors, I believe that after correcting the comments provided by the reviewers, the article is of better quality, so I approve it for publication in the journal. Congratulations.

Author Response

Round 2 Responses to Reviewers

Manuscript: antibiotics-3651380 “Metabolomic Analysis of Terminalia citrina Leaf Extracts That Inhibit the Growth of Gastrointestinal Bacterial Pathogens, Including Methicillin-Resistant Staphylococcus aureus

Reviewer 4

Dear authors, I believe that after correcting the comments provided by the reviewers, the article is of better quality, so I approve it for publication in the journal. Congratulations.

We are grateful to the reviewer for this positive response.

Round 3

Reviewer 3 Report

Comments and Suggestions for Authors

In this revision, the authors  have taken all my comments into consideration and have significantly improved the manuscript. The work is clearer. The results and discussion are no longer confusing. Therefore, I recommend publication of this work in this form.